# IMPROVED STATISTICAL AND COMPUTATIONAL COMPLEXITY OF THE MEAN-FIELD LANGEVIN DYNAMICS UNDER STRUCTURED DATA

[*]**Atsushi Nitanda**[1,2], **Kazusato Oko**[3,4], **Taiji Suzuki**[3,4], **Denny Wu**[5,6]

[*]Alphabetical ordering.   [1]IHPC, Agency for Science, Technology and Research, Singapore,
[2]CFAR, Agency for Science, Technology and Research, Singapore,
[3]Department of Mathematical Informatics, the University of Tokyo,
[4]Center for Advanced Intelligence Project, RIKEN,   [5]Center for Data Science, New York University,
[6]Center for Computational Mathematics, Flatiron Institute

`atsushi_nitanda@cfar.a-star.edu.sg`, `oko-kazusato@g.ecc.u-tokyo.ac.jp`,
`taiji@mist.i.u-tokyo.ac.jp`, `dennywu@nyu.edu`

## ABSTRACT

Recent works have shown that neural networks optimized by gradient-based methods can adapt to sparse or low-dimensional target functions through feature learning; an often studied target is the sparse parity function on the unit hypercube. However, such isotropic data setting does not capture the anisotropy and low intrinsic dimensionality exhibited in realistic datasets. In this work, we address this shortcoming by studying how gradient-based feature learning interacts with structured (anisotropic) input data: we consider the classification of $k$-sparse parity on high-dimensional orthotope where the feature coordinates have varying magnitudes, and analyze the learning complexity of the mean-field Langevin dynamics (MFLD), which describes the noisy gradient descent update on two-layer neural network. We show that the statistical complexity (i.e. sample size) and computational complexity (i.e. network width) of MFLD can both be improved when prominent directions of the anisotropic input data align with the support of the target function. Moreover, by employing a coordinate transform determined by the gradient covariance, the width can be made independent of the target degree $k$. Lastly, we demonstrate the benefit of feature learning by establishing a kernel lower bound on the classification error, which applies to neural networks in the lazy regime.

## 1 INTRODUCTION

We consider the learning of a two-layer nonlinear neural network (NN) with $N$ neurons:

$$f(z) = \frac{1}{N} \sum_{i=1}^{N} h_{x^{(i)}}(z), \quad z \in \mathbb{R}^d, \ h_{x^{(i)}}(z) : \mathbb{R}^d \to \mathbb{R}, \tag{1}$$

where $h_{x^{(i)}}(z)$ is one neuron with trainable parameter $x^{(i)}$, e.g., we may set $x^{(i)} \in \mathbb{R}^d$ and $h_{x^{(i)}}(z) = \sigma(\langle z, x^{(i)} \rangle)$ with some nonlinearity $\sigma : \mathbb{R} \to \mathbb{R}$. One crucial benefit of the model (1) is the ability to learn representation that adapts to the learning problem, such as sparsity and low-dimensional structures. Indeed, it has been shown that this *feature learning* ability enables NNs trained with gradient-based algorithms to avoid the curse of dimensionality and outperform non-adaptive methods such as kernel models in learning various low-dimensional target functions (Abbe et al., 2022; Ba et al., 2022; Damian et al., 2022; Bietti et al., 2022; Mousavi-Hosseini et al., 2022; Abbe et al., 2023).

A noticeable example of low-dimensional problem is the classification of $k$-sparse parity, where the target label is defined as the sign of the product of $k \ll d$ coordinates: $f_*(z_i) = \text{sign}\left(\prod_{i=1}^{k} z_i\right)$, where $z_i$ denotes the $i$-th coordinate of vector $z$. The classical XOR problem corresponds to the case where $k = 2$ and input on the unit hypercube. Efficiently learning this target function requires the first-layer parameters of the NN to identify the relevant $k$-dimensional subspace, which can be achieved via gradient-based feature learning (Daniely and Malach, 2020; Refinetti et al., 2021; Frei et al., 2022; Barak et al., 2022; Ben Arous et al., 2022).

One particularly relevant feature learning paradigm for the parity problem is the mean-field analysis, which lifts the optimization problem into the space of measures (Nitanda and Suzuki, 2017; Chizat and Bach, 2018; Mei et al., 2018; Rotskoff and Vanden-Eijnden, 2018). For isotropic input ($z_i \in \{-1, +1\}$), mean-field NN can learn the parity function with *linear sample complexity*. Specifically, Wei et al. (2019); Chizat and Bach (2020); Telgarsky (2023) proved a $\mathcal{O}(d/n)$ classification error for 2-parity (XOR), when the NN is optimized by (modified) gradient flow. Very recently, Suzuki et al. (2023b) considered a noisy variant of gradient descent termed the *mean-field Langevin dynamics* (MFLD), and showed that the $\mathcal{O}(d/n)$ rate remains valid for the isotropic $k$-parity problem with dimension-free $k$. While the computational complexity is demanding due to the exponential width required in the mean-field analysis, one remarkable feature is the statistical complexity *decouples* the degree $k$ from the exponent in the dimension dependence; this contrasts the NTK analysis where a sample size of $n = \Omega(d^k)$ is typically needed to learn a degree-$k$ polynomial on isotropic input data (Ghorbani et al., 2019; Mei et al., 2022), and thus demonstrates the benefit of feature learning.

**Feature learning under structured data.** Noticeably, most existing analyses on the parity problem are restricted to the *isotropic* setting, where the input features do not provide any information of the support of the target function. On the other hand, realistic datasets are often structured, and different feature directions may have different magnitudes that guide the algorithm towards efficient learning. For example, real-world data often has low intrinsic dimensionality (Fodor, 2002; Pope et al., 2021), and the observation that input directions with larger variation tend to have good predictive power has motivated various data preprocessing procedures such as PCA (Hastie et al., 2009).

Recent works have indeed illustrated that in certain regression settings with low-dimensional target, anisotropic input data can improve the performance of both kernel methods and NNs. However, these results either did not take into account the optimization dynamics of NN (Suzuki and Nitanda, 2019; Ghorbani et al., 2020), or characterized the feature learning dynamics in a "narrow-width" setting (Ba et al., 2023; Mousavi-Hosseini et al., 2023) which differs from the mean-field regime. Moreover, classification and regression problems have fundamentally different structures, and thus existing regression analyses do not directly translate to the $k$-parity classification problem. Therefore, our goal is to investigate the interplay between structured data and feature learning in the problem setting of classifying $k$-sparse parity function on *anisotropic* input data with mean-field NN.

## 1.1 OUR CONTRIBUTIONS

We study the statistical and computational complexity of the mean-field Langevin dynamics in learning a $k$-sparse parity target function on anisotropic input data. Specifically, we consider the following generating process of the data-label pairs $(z, y)$,

$$z = A\tilde{z}, \quad y = \text{sign}\big( \prod_{i \in I_k} \tilde{z}_i \big), \quad \text{where } \tilde{z}_i \overset{i.i.d.}{\sim} \text{Unif}\big( \{-1/\sqrt{d}, 1/\sqrt{d}\} \big),$$

for some matrix $A \in \mathbb{R}^{d \times d}$ which controls the anisotropy. Extending the convergence analysis of MFLD in Suzuki et al. (2023a;b), we prove discrete-time and finite-width learning guarantees for two-layer neural network optimized with noisy gradient descent for this general data setting. We then specialize our general learnability result to specific examples where MFLD benefits from structured data through an improvement of the constant in the *logarithmic Sobolev inequality* of a proximal Gibbs measure associated with the training dynamics, and demonstrate that both the statistical and computational complexity improves upon that in the isotropic setting. In particular, we show that

- When the feature directions of $z$ with large magnitude align with the support of the target function $I_k$, then MFLD can achieve better statistical complexity (required sample size) and computational complexity (required network width) compared to the isotropic setting in Suzuki et al. (2023b).

- If we apply a coordinate transformation based on the gradient covariance matrix, then the required width for MFLD can be made independent of the degree $k$. This is equivalent to an anisotropic $\ell_2$ regularization, and we prove that the weighting matrix can be estimated from the first gradient step.

- We also provide a classification error lower bound for kernel methods in the anisotropic parity setting, which highlights the advantage of gradient-based feature learning via MFLD.

In Table 1 we summarize and compare our results against prior works on learning sparse parity functions. To clearly illustrate the improved dimension dependence, we state our rates for a simple spiked covariance model analogous to the setting in Ghorbani et al. (2020); Ba et al. (2023) (see

| data | result type | regime/method | sample size | width | iterations | authors |
|------|------------|---------------|-------------|-------|------------|---------|
| Isotropic | upper bound | NTK/SGD | $d^2/\epsilon$ | $d^8$ | $d^2/\epsilon$ | Ji and Telgarsky (2019) |
| | | two-phase SGD | $d^{k+1}/\epsilon^2$ | $\mathcal{O}(1)$ | $d/\epsilon^2$ | Barak et al. (2022) |
| | | mean-field/GF | $d/\epsilon$ | $\infty$ | $\infty$ | Wei et al. (2019) |
| | | mean-field/GF | $d/\epsilon$ | $d^{d/2}$ | $\infty$ | Telgarsky (2023) |
| | | MFLD | $d/\epsilon$ | $\exp(d)$ | $\exp(d)$ | Suzuki et al. (2023b) |
| | lower bound | random features | – | $d^k$ | – | Barak et al. (2022) |
| Anisotropic | upper bound | MFLD | $d_{\text{eff}}/\epsilon$ | $\exp(d_{\text{eff}})$ | $\exp(d_{\text{eff}})$ | Theorem 1 |
| | | MFLD (transformed) | $d_{\text{eff}}^k d + 1/\epsilon$ | $d^3$ | $\mathcal{O}(1)$ | Theorem 2 |
| | lower bound | kernel | $d_{\text{eff}}^k$ | – | – | Theorem 3 |

Table 1: Statistical and computational complexity for the $k$-sparse parity classification, omitting polylogarithmic terms. $d$ is the dimensionality, and $n$ is the sample size. We set $y = \text{sign}\left(\prod_{i=1}^k \tilde{z}_i\right)$, $A = \text{diag}(s_1, s_2, ..., s_d)$, where $s_1 = ... = s_k = d^{\alpha/2}$ for $\alpha \geq 0$, and $s_{k+1} = ... = s_d = 1$; following Ghorbani et al. (2020) we define the *effective dimension* as $d_{\text{eff}} := d^{1-\alpha} \ll d$ when $\alpha > 0$. We note that results from Wei et al. (2019); Telgarsky (2023) do not cover the general $k$-parity setting, so we state the complexity for 2-parity (XOR). For the lower bound, we restate (Barak et al., 2022, Theorem 5) for bounded norm random features predictor. Finally, for the kernel lower bound in Theorem 3 we only track the dimension dependence.

(10) in Section 4.1): the data-label pairs $(z, y)$ are generated as $y = \text{sign}\left(\prod_{i=1}^k z_i\right)$ for $k = \mathcal{O}_d(1)$, where the informative directions are given as $z_i \in \{\pm d^{(\alpha-1)/2}\}$ $(i = 1, \cdots, k)$ for $\alpha \in [0, 1]$, and the uninformative directions $z_i \in \{\pm d^{-1/2}\}$ $(i = k + 1, \cdots, d)$. In this example, larger $\alpha$ (hence smaller $d_{\text{eff}} = d^{1-\alpha}$) corresponds to stronger anisotropy, which facilities feature learning due to the alignment between the low-dimensional structure and the target function. As shown in Table 1, this benefit is evident in both the original MFLD and the coordinate-transformed version.

## 2 PROBLEM SETTING

**$k$-sparse parity classification.** We consider the following binary classification problem.

**Definition 1** ($k$-sparse parity problem under linear transformation)**.** *Given invertible matrix A, the input random variable Z and the corresponding label Y are generated as*

$$Z = A\tilde{Z}, \quad Y = \text{sign}\left(\prod_{i \in I_k} \tilde{Z}_i\right),$$

*where $\tilde{Z}$ is distributed from the uniform distribution on $\{\pm 1/\sqrt{d}\}^d$, and $\|Z\| \leq 1$ almost surely.*

This definition includes the well-studied XOR (Wei et al., 2019; Telgarsky, 2023) as a special case.

**Example 1** (Isotropic XOR)**.** *We take $A = I_d$ and $Y = \text{sign}(\tilde{Z}_1 \tilde{Z}_2)$ ($k = 2$).*

Similarly, we can also cover $k$-parity on isotropic data (Barak et al., 2022; Suzuki et al., 2023b).

As for anisotropic data, an example that we will consider in the subsequent sections is the following axis-aligned setting, where the coordinates are independent but may have different magnitudes.

**Example 2** (Axis-aligned anisotropic $k$ parity)**.** *There exist positive reals $s_i > 0$ $(i = 1, \ldots, d)$ such that the support of $P_Z$ (the distribution of Z) is given by $\mathcal{S} := \{\pm s_1\} \times \{\pm s_2\} \times \cdots \times \{\pm s_d\}$ where $\sum_{j=1}^d s_j^2 = 1$, and $(z_i)_{i=1}^d$ are mutually independent and $P(z_i = s_i) = P(z_i = -s_i) = 1/2$. The $k$-sparse parity label corresponds to the sign of the product of $k$-indices $I_k \subset \{1, \ldots, d\}$.*

**Mean-field two-layer network.** Let $h_x(\cdot) : \mathbb{R}^d \to \mathbb{R}$ be one neuron associated with parameter $x = (x_1, x_2, x_3) \in \mathbb{R}^{d+1+1}$ in a two-layer neural network: given an input $z \in \mathbb{R}^d$,

$$h_x(z) = \bar{R}[\tanh(z^\top x_1 + x_2) + 2\tanh(x_3)]/3, \tag{2}$$

where $\bar{R} \in \mathbb{R}$ is an output scale of the network and an extra $\tanh$ activation for the bias term $x_3 \in \mathbb{R}$ is placed to make the function bounded following Suzuki et al. (2023b). Let $\mathcal{P}$ be the set of Borel probability measures on $\mathbb{R}^{\bar{d}}$ where $\bar{d} = d + 2$ and $\mathcal{P}_p$ be the subset of $\mathcal{P}$ with finite $p$-th moment: $\mathbb{E}_\mu[\|X\|^p] < \infty$ $(\mu \in \mathcal{P})$. The mean-field neural network is defined by integrating infinitely many neurons $h_x$ over $\mathbb{R}^{\bar{d}}$ with the distribution $\mu \in \mathcal{P}$: $f_\mu(\cdot) = \int h_x(\cdot)\mu(\text{d}x)$.

Let $\ell(\cdot, \cdot) : \mathbb{R} \times \mathbb{R} \to \mathbb{R}_{\geq 0}$ be a smooth and convex loss function for the binary classification. Typically, we consider the logistic loss function $\ell(f, y) = \log(1 + \exp(-yf))$ where $f \in \mathbb{R}$, $y \in \{\pm 1\}$. We

also denote $\ell(yf) = \ell(f, y)$ Then, the empirical risk and the population risk of $f_\mu$ are defined as

$$L(\mu) := \frac{1}{n} \sum_{i=1}^{n} \ell(y_i f_\mu(z^{(i)})), \quad \bar{L}(\mu) := \mathbb{E}[\ell(Y f_\mu(Z))].$$

To avoid overfitting, we consider a regularized empirical risk $F(\mu) := L(\mu) + \lambda \mathbb{E}_{X \sim \mu}[\lambda_1 \|X\|^2]$ with the regularization parameters $\lambda, \lambda_1 \geq 0$. In addition, we introduce the entropy regularized risk:

$$\mathcal{L}(\mu) = F(\mu) + \lambda \text{Ent}(\mu). \tag{3}$$

We can immediately see that $\mathcal{L}$ is equivalent to $L(\mu) + \lambda \text{KL}(\nu, \mu)$ up to constant, where $\text{KL}(\nu, \mu) = \int \log(\mu/\nu) \mathrm{d}\mu$ is the KL divergence between $\nu$ and $\mu$, and $\nu$ is the Gaussian distribution with mean 0 and variance $I/(2\lambda_1)$. A remarkable advantage of mean-field parameterization is that the above objectives become convex functional with respect to the distribution $\mu$, since $\mu$ linearly acts on $f_\mu$.

## 3 MEAN-FIELD LANGEVIN DYNAMICS

The mean-field Langevin dynamics is defined by the following stochastic differential equation:

$$\mathrm{d}X_t = -\nabla \frac{\delta F(\mu_t)}{\delta \mu}(X_t) \mathrm{d}t + \sqrt{2\lambda} \mathrm{d}W_t, \quad \mu_t = \text{Law}(X_t), \tag{4}$$

starting at $X_0 \sim \mu_0$, where $(W_t)_{t \geq 0}$ is the $d$-dimensional standard Brownian motion, and $\frac{\delta F(\mu_t)}{\delta \mu}$ is the first variation of $F$, which, in our setting, is written as $\frac{\delta F(\mu)}{\delta \mu}(x) = \frac{1}{n} \sum_{i=1}^{n} \ell'(y_i f_\mu(z^{(i)})) y_i h_x(z_i) + \lambda(\lambda_1 \|x\|^2)$. The Fokker-Planck equation of SDE (4) is given by[1]

$$\partial_t \mu_t = \lambda \Delta \mu_t + \nabla \cdot \left[ \mu_t \nabla \frac{\delta F(\mu_t)}{\delta \nu} \right] = \nabla \cdot \left[ \mu_t \nabla \left( \lambda \log(\mu_t) + \frac{\delta F(\mu_t)}{\delta \nu} \right) \right]. \tag{5}$$

Several studies (Mei et al., 2018; Hu et al., 2019) showed that MFLD (4) globally optimizes the objective (3), that is, when $t \to \infty$ we have $\mathcal{L}(\mu_t) \to \mathcal{L}(\mu_{[\lambda]})$, where $\mu_{[\lambda]} := \text{argmin}_{\mu \in \mathcal{P}} \mathcal{L}(\mu)$.

For a practical algorithm, we need to consider a space- and time-discretized version of the MFLD, that is, we approximate the solution $\mu_t$ by an empirical measure $\mu_{\mathscr{X}} = \frac{1}{N} \sum_{i=1}^{N} \delta_{X_i}$ corresponding to a set of finite particles $\mathscr{X} = (X^i)_{i=1}^{N} \subset \mathbb{R}^{\bar{d}}$. Let $\mathscr{X}_\tau = (X_\tau^i)_{i=1}^{N} \subset \mathbb{R}^{\bar{d}}$ be $N$ particles at the $\tau$-th update ($\tau \in \{0, 1, 2, \dots\}$), and define $\mu_\tau = \mu_{\mathscr{X}_\tau}$ as a finite particle approximation of the population counterpart. Then, the discretized MFLD is defined as follows: $X_0^i \sim \mu_0$, and $\mathscr{X}_\tau$ is updated as

$$X_{\tau+1}^i = X_\tau^i - \eta \nabla \frac{\delta F(\mu_\tau)}{\delta \mu}(X_\tau^i) + \sqrt{2\lambda\eta} \xi_\tau^i, \tag{6}$$

where $\eta > 0$ is the step size, $\xi_\tau^i \sim_{i.i.d.} N(0, I)$. Note that in the context of mean-field neural network (1), the discretized update (6) simply corresponds to the noisy gradient descent algorithm, where a Gaussian perturbation is added at each gradient step. We write $f_{\mathscr{X}} := f_{\mu_{\mathscr{X}}}$ for simplicity of notation.

### 3.1 LOGARITHMIC SOBOLEV INEQUALITY

Nitanda et al. (2022); Chizat (2022) have established the exponential convergence of MFLD by exploiting the *proximal Gibbs distribution* $p_\mu$ associated with $\mu \in \mathcal{P}$. The density of $p_\mu$ is given by

$$p_\mu(X) \propto \exp\left( -\frac{1}{\lambda} \frac{\delta F(\mu)}{\delta \mu}(X) \right).$$

The smoothness of the loss function and the $\tanh$ activation guarantee the existence of the unique minimizer $\mu^*$ of $\mathcal{L}$, which also solves the equation: $\mu = p_\mu$ (see Proposition 2.5 of Hu et al. (2019)). The key in their proofs is to show a *logarithmic Sobolev inequality* (LSI) on the Gibbs measure $p_\mu$.

**Definition 2** (Logarithmic Sobolev inequality). *Let $\mu$ be a Borel probability measure on $\mathbb{R}^d$. We say $\mu$ satisfies the LSI with a constant $\alpha > 0$ if for any smooth function $\phi : \mathbb{R}^d \to \mathbb{R}$ with $\mathbb{E}_\mu[\phi^2] < \infty$,*

$$\mathbb{E}_\mu[\phi^2 \log(\phi^2)] - \mathbb{E}_\mu[\phi^2] \log(\mathbb{E}_\mu[\phi^2]) \leq \frac{2}{\alpha} \mathbb{E}_\mu[\|\nabla \phi\|_2^2].$$

We can apply the classical Bakry-Emery and Holley-Strook arguments (Bakry and Émery, 1985; Holley and Stroock, 1987) to derive the LSI constant on the Gibbs distribution whose potential is the sum of the strongly convex function and bounded function. In particular, if $\|\frac{\delta L(\mu)}{\delta \mu}\|_\infty \leq B$, we can establish the LSI for the proximal Gibbs distribution with $\alpha \geq \lambda_1 \exp(-4B/\lambda)$. In our case, since the logistic loss is employed and each neuron $h_x$ is bounded by $\bar{R}$, we have $B = \bar{R}$ and therefore

$$\alpha \geq \lambda_1 \exp(-4\bar{R}/\lambda). \tag{7}$$

---

[1]This should be interpreted in a weak sense, that is, for any continuously differentiable function $\phi$ with a compact support, $\int \phi \mathrm{d}\mu_t - \int \phi \mathrm{d}\mu_s = -\int_s^t \int \nabla \phi \cdot (\nabla \log(\mu_t) - \nabla \frac{\delta F(\mu_t)}{\delta \nu}) \mathrm{d}\mu_\tau \mathrm{d}\tau$.

## 3.2 QUANTITATIVE ANALYSIS OF MFLD

**Convergence guarantee.** As shown in Chen et al. (2022); Suzuki et al. (2022), the LSI constant determines not only the convergence rate of the continuous dynamics, but also the number of particles (i.e., width of the neural network) to approximate the mean-field limit. Let us consider the linear functional of a distribution $\mu^{(N)}$ of $N$ particles $\mathscr{X} = (X^i)_{i=1}^N \subset \mathbb{R}^{\bar{d}}$ defined by

$$\mathcal{L}^N(\mu^{(N)}) = N\mathbb{E}_{\mathscr{X} \sim \mu^{(N)}}[F(\mu_{\mathscr{X}})] + \lambda\mathrm{Ent}(\mu^{(N)}).$$

Let $\mu_\tau^{(N)}$ be the distribution of particles $\mathscr{X}_\tau = (X_\tau^i)_{i=1}^N$ at the $\tau$-th iteration, and define $\Delta_\tau = \frac{1}{N}\mathcal{L}^N(\mu_\tau^{(N)}) - \mathcal{L}(\mu_{[\lambda]})$. Suzuki et al. (2023a) established the convergence rate of MFLD as follows.

**Proposition 1.** *Let $\bar{B}^2 := \mathbb{E}[\|X_0^i\|^2] + \frac{1}{\lambda\lambda_1}\left[\left(\frac{1}{4} + \frac{1}{\lambda\lambda_1}\right)\bar{R}^2 + \lambda d\right]$ and $\delta_\eta := C_1\bar{L}^2(\eta^2 + \lambda\eta)$, where $\bar{L} = 2\bar{R} + \lambda\lambda_1$ and $C_1 = 8(\bar{R}^2 + \lambda\lambda_1\bar{B}^2 + d) = O(d + \lambda^{-1})$. Then, if $\lambda\alpha\eta \leq 1/4$ and $\eta \leq 1/4$, then the neural network trained by MFLD converges to the optimal network $f_{[\lambda]}$ as*

$$\mathbb{E}_{\mathscr{X}_\tau \sim \mu_\tau^{(N)}}\left[\sup_{z \in \mathrm{supp}(P_Z)}(f_{\mathscr{X}_\tau}(z) - f_{\mu_{[\lambda]}}(z))^2\right] \leq \frac{4\bar{L}^2}{\lambda\alpha}\Delta_\tau + \frac{2}{N}\bar{R}^2,$$

*where $\Delta_\tau$ is further bounded by $\Delta_\tau \leq \exp\left(-\lambda\alpha\eta\tau/2\right)\Delta_0 + \frac{2}{\lambda\alpha}\bar{L}^2C_1\left(\lambda\eta + \eta^2\right) + \frac{4C_\lambda}{\lambda\alpha N}$.*

In particular, for a given $\epsilon^* > 0$, the right hand side can be bounded by $\epsilon^* + \frac{2\bar{R}^2}{N}$ after $T = O\left(\frac{1}{\lambda\alpha\eta}\log(1/\epsilon^*)\right)$ iterations with the step size $\eta = O(\lambda\alpha^2\epsilon^*/C_1 + \lambda\alpha\sqrt{\epsilon^*/C_1})$. In terms of generalization error (Proposition 2), the optimization error can be set as $\epsilon^* = O(1/(n\lambda)^2)$. Then, the required total number of iteration $T$ and the number of particles $N$ can be bounded by

$$T \leq O\left((d + \lambda^{-1})n^2\exp(16\bar{R}/\lambda)\log(n\lambda)\right), \quad N \leq O((\epsilon^*\lambda\alpha)^{-2}) = O\left(n^2\exp(8\bar{R}/\lambda)\right). \quad (8)$$

From this evaluation, we see that it is crucial to select the regularization strength $\lambda$ so that the loss is sufficiently small. In the following section, we investigate how structured data affects the choice of $\lambda$.

**Generalization error bound.** Now we state the classification error bound of the neural network optimized by MFLD. For this purpose, we introduce the following assumption which will be verified later on for the anisotropic parity setting.

**Assumption 1.** *There exists $c_0 > 0$ and $R > 0$ such that the following conditions are satisfied:*

- *There exists $\mu^* \in \mathcal{P}$ such that $\mathrm{KL}(\nu\|\mu^*) \leq R$ and $L(\mu^*) \leq \ell(0) - c_0$.*
- *For any $\lambda < c_0/R$, the risk minimizer $\mu_{[\lambda]}$ of $\mathcal{L}(\mu)$ satisfies $Yf_{\mu_{[\lambda]}}(X) \geq c_0$ almost surely.*

Here $c_0$ characterizes the margin of a solution $\mu^*$ and $R$ controls "difficulty" of the problem. Indeed, if larger $R$ is required, the Bayes optimal solution should be far away from the prior $\nu$. Hence, we expect that obtaining a good classifier is more difficult. Let $\hat{\mu}$ be an approximately optimal solution of $\mathcal{L}$ with $\epsilon^*$ accuracy: $\mathcal{L}(\hat{\mu}) \leq \min_{\mu \in \mathcal{P}} \mathcal{L}(\mu) + \epsilon^*$; we have the following generalization error bounds.

**Proposition 2** (Suzuki et al. (2023b)). *Let $M_0 = (\epsilon^* + 2(\bar{R} + 1))/\lambda$ and suppose that $\lambda < c_0/R$.*

*(i) If the sample size $n$ satisfies*

$$n > C\frac{\bar{R}^2}{c_0^2\lambda^2}\left[\lambda\left(\bar{R} + \frac{\lambda}{\bar{R}^2 n}\right) + \bar{R}^2(1 + \log\log_2(n^2M_0\bar{R})) + n\lambda\epsilon^*\right] =: S,$$

*with an absolute constant $C$, then $f_{\hat{\mu}}$ satisfies $P\left(Yf_{\hat{\mu}}(Z) \leq 0\right) = 0$ (the Bayes optimal classifier) with probability $1 - \exp(-\frac{n\lambda^2}{32\bar{R}^4}(c_0^2 - S/n))$.*

*(ii) When the sample size does not satisfy the condition $n > S$, we still have an alternative error bound: there exists an absolute constant $C > 0$ such that*

$$P(Yf_{\hat{\mu}}(Z) \leq 0) \leq C\beta(c_0)\left[\frac{\bar{R}^2}{n\lambda}\left(1 + t + \log\log_2(n^2M_0\bar{R})\right) + \frac{1}{n}\left(\bar{R} + \frac{\lambda}{\bar{R}^2 n}\right) + \epsilon^*\right],$$

*with probability $1 - \exp(-t)$, where $\beta(c_0) := 1/[\ell(0) - (\ell(c_0) - c_0\ell'(c_0))]$.*

This result states that if we take the regularization parameter $\lambda$ sufficiently small as $\lambda < \mathcal{O}(1/R)$, then for sufficiently large sample size such that $n > S = \Omega(1/\lambda^2)$, we have an exponential convergence of the expected classification error as $\mathbb{E}_{D^n}[P(Yf_{\hat{\mu}}(Z) \leq 0)] \leq \exp(-\Omega(n\lambda^2))$; otherwise, we sill have a linear decay $\mathbb{E}_{D^n}[P(Yf_{\hat{\mu}}(Z) \leq 0)] = \mathcal{O}(1/(n\lambda))$. Hence, the classification error and its convergence rate is almost completely characterized by $R$ through the choice of $\lambda = \mathcal{O}(1/R)$: for a problem with large $R$, we need to pay greater sample complexity.

It is also worth noting that the value of $R$ affects not only the statistical complexity but also the computational complexity. Remember that the number of iterations $T$ and the network width $N$ also depend on $\lambda$ through Eq. (8). Indeed, by taking $\lambda = c_0/R$, we arrive at $T = \mathcal{O}(\exp(16\bar{R}R/c_0)\log(n))$ and $N = \mathcal{O}(\exp(8\bar{R}R/c_0))$, which has exponential dependence on $R$.

Therefore, the goal of the subsequent sections is to answer the following question in the affirmative:

*Can we utilize the anisotropy of input data to reduce the value of $R$,*
*hence improving the statistical and computational complexity of MFLD?*

## 4 MAIN RESULT: LEARNING UNDER STRUCTURED DATA

### 4.1 STATISTICAL AND COMPUTATIONAL COMPLEXITY FOR ANISOTROPIC DATA

Now we analyze how the anisotropic property of the input affects the generalization error and the computational complexity through the aforementioned measure of problem difficulty $R$. We first present a framework for the general problem setting in Definition 1. Let $\tilde{\phi} = (\tilde{\phi}_1, \ldots, \tilde{\phi}_d)^\top \in \mathbb{R}^d$ as

$$\tilde{\phi}_i = \begin{cases} \sqrt{d} & (i \in I_k), \\ 0 & (i \notin I_k). \end{cases} \tag{9}$$

Then, we have the following proposition that controls $R$ in terms of the transformation matrix $A$.

**Proposition 3.** *Define $\phi := A^{-1}\tilde{\phi}$ where $\tilde{\phi}$ is defined by Eq. (9). For $\bar{R} = k$, there exists $\mu^* \in \mathcal{P}$ and $R$ such that*
$$\mathrm{KL}(\nu || \mu^*) \leq R = c_1(\|\phi\|^2 + k^2)\log(k)^2,$$
*and $L(\mu^*) \leq \ell(0) - c_2$, where $c_1, c_2 > 0$ are absolute constants.*

Under the conditions in this proposition, we can show that the minimizer of the MFLD objective achieves the Bayes optimal classifier with a positive margin as follows.

**Proposition 4.** *Assume that there exists $\mu^* \in \mathcal{P}$ such that the conditions in Proposition 3 is satisfied with $R$ and $\bar{R}$ in the statement. Then, if we choose the regulaization parameter $\lambda$ as $\lambda < c_2/(2R)$, then the minimizer $\mu_{[\lambda]}$ of the MFLD objective satisfies*
$$\max\{\bar{L}(\mu_{[\lambda]}), L(\mu_{[\lambda]})\} < \ell(0) - \frac{c_2}{2},$$
*and $f_{\mu_{[\lambda]}}$ is a perfect classifier with margin $c_2$, i.e., $Yf_{\mu_{[\lambda]}}(Z) \geq \frac{c_2}{2}$ almost surely.*

The proofs of both propositions can be found in Appendix A in the appendix. These general results state that Assumption 1 is satisfied for the general problem setting in Definition 1. Now we consider special cases where concrete sample complexity and computational complexity can be derived. For example, we have the following evaluation for the $k$-sparse parity with anisotropic covariance.

**Example: Anisotropic $k$-sparse parity.** In the $k$-parity setting (Example 2), Assumption 1 is satisfied with constants specified in the following propositions, which follow from Proposition 4.

**Corollary 1** (Anisotropic $k$-sparse parity). *Suppose that $(Z, Y)$ is generated from the anisotropic $k$ parity problem (Example 2). Then, for $\bar{R} = k$, there exists $\mu^* \in \mathcal{P}$ satisfying $\mathrm{KL}(\nu || \mu^*) \leq R$ where*

$$R = c_1\left(\sum_{i \in I_k} s_i^{-2}\right)\log(k)^2,$$

*and $L(\mu^*) \leq \ell(0) - c_2$, where $c_1, c_2 > 0$ are absolute constants.*

This result highlights the benefit of structured data. Observe that isotropic covariance corresponds to $s_i = 1/\sqrt{d}$ $(i = 1, \ldots, d)$, where $R$ needs to be $\tilde{\mathcal{O}}(kd)$, which then leads to exponential dimension dependency in the computational complexity, and also dimension-dependent sample complexity, as shown in Suzuki et al. (2023b). On the other hand, if the input covariance is anisotropic so that $s_j^2 > \Omega(1/k)$ for $j \in I_k$ (i.e., the input $Z_j$ is large for the informative coordinates $j \in I_k$ and other coordinates are small), then the value of $R$ becomes dimension-free: $R = \mathcal{O}(k^2 \log(k)^2)$.

Substituting the values of $R$ and $\bar{R}$ to the generalization error and computational complexity bounds, we obtain the following corollary.

**Theorem 1** ($k$-sparse parity setting). *Define $S_{I_k}^2 := \sum_{j \in I_k} s_j^{-2}$. Under the same setup as Corollary 1, we may take $R = O(S_{I_k}^2 \log(k)^2)$, $\bar{R} = k$ and $\lambda = O(1/R) = O(1/(S_{I_k}^2 \log(k)^2))$ so that the classification error is bounded by*

$$P(Y f_{\mu_{[\lambda]}} < 0) \leq O\left( \frac{k S_{I_k}^2 \log(k)^2}{n} (\log(1/\delta) + \log\log(n)) \right),$$

*with probability $1 - \delta$. Moreover, if $n = \Omega(k^4 S_{I_k}^4 \log(k)^4)$, then $P(Y f_{\mu_{[\lambda]}} \leq 0) = 0$ with probability*

$$1 - \exp[-\Omega(n/(k^4 S_{I_k}^4 \log(k)^4))].$$

*For the computational cost, it suffices to take the number of iterations $T$ and network width $N$ as*

$$T = O(S_{I_k}^2 \log(k)^2 n \log(nd) \exp[O(k S_{I_k}^2 \log(k)^2)]), \quad N = O(n^2 \exp(O(k S_{I_k}^2 \log(k)^2)))),$$

*respectively, to achieve the same statistical complexity as described above.*

As mentioned above, for sufficiently anisotropic data such that $S_{I_k}^2 = k^2$, the computational complexity becomes completely polynomial order with respect to the dimension $d$; this is in stark contrast to the isotropic setting, where the complexity has exponential order with respect to $d$.

Now we provide two examples of the $k$-parity problem in Example 2 (i.e., $I_k = \{1, \ldots, k\}$) where covariance structure allows us to smoothly interpolate between the isotropic and anisotropic setting.

- *Power-law decay.* We set

$$s_i^2 = c_d i^{-\alpha}, c_d = \Theta(d^{1-\alpha}), \quad \text{where } \alpha \in [0, 1).$$

We have that $S_{I_k}^2 = \mathcal{O}(d^{1-\alpha})$ leading to $R = \mathcal{O}(k^{1+\alpha} d^{1-\alpha} \log(k)^2)$. This interpolates between the isotropic and the completely anisotropic setting $S_{I_k}^2 = k^2$ by adjusting $\alpha$ between $(0, 1)$.

- *Spiked covariance.* Similar to Ghorbani et al. (2020); Ba et al. (2023), we set

$$s_i^2 = \Theta(d^{\alpha - 1}) \text{ for } i \in I_k, \quad s_i = \Theta(d^{-1}) \text{ otherwise}, \quad \text{where } \alpha \in [0, 1]. \tag{10}$$

In this case we have $R = k d^{1-\alpha} \log(k)^2$, which becomes dimension-free when $\alpha$ approaches 1.

**Remark 1.** *For the spiked covariance setting above, the computational and statistical complexity of MFLD is governed by the effective dimensions $d_{eff} = d^{1-\alpha}$ defined in Ghorbani et al. (2020). As the input becomes more anisotropic, $d_{eff}$ decreases and hence the learning problem becomes easier.*

## 4.2 ENHANCING ANISOTROPY VIA COORDINATE TRANSFORM

From the previous analysis, we see that anisotropic data can indeed improve both the statistical and computational complexity. This being said, it is worth noting that unless the problem is sufficiently anisotropic such that $R$ becomes cost, the computational cost would still be super-polynomial in terms of dimension dependence. The goal of this section is to show that the computational complexity can be further improved by exploiting the anisotropy of the learning problem. Specifically, we utilize the gradient covariance matrix to estimate the informative subspace, similar to the one-step gradient feature learning procedure studied in Ba et al. (2022); Damian et al. (2022); Barak et al. (2022).

Let $\sigma(w^\top z) = h_x(z)$ for $(x_1, x_2, x_3) = (w, b_1, b_2)$ for fixed $b_1$ and $b_2$. We initialize the particles $\mathcal{X}_0 = \{(w_l, b_1, b_2)\}_{l=1}^{N/2} \cup \{(-w_l, -b_1, -b_2)\}_{l=1}^{N/2}$ by generating $w_l$ from the uniform distribution $\mathcal{U}(\mathcal{B}_{c_0})$ on the ball with sufficiently small radius $c_0 > 0$. The gradient for each neuron is given as

$$g(w_l) = \frac{1}{n} \sum_{i=1}^{n} \ell'(y_i f_{\mathcal{X}_0}(z^{(i)})) y_i z \sigma'(w_l^\top z^{(i)}).$$

Note that we have $f_{\mathscr{X}_0}(Z) = 0$ almost surely. We then calculate the covariance as

$$G = \tfrac{1}{N} \sum_{l=1}^{N} g(w_l) g(w_l)^\top,$$

to estimate the informative subspace. Define the "regularized covariance" $\hat{G} = G + \hat{\lambda}_0 I$. For this choice of $\hat{G}$, we apply the following coordinate transform to the input $Z$:

$$\hat{Z} \leftarrow c_A \hat{G}^{1/2} Z,$$

where $c_A$ is a scaling parameter so that $\|\hat{Z}\| \leq 1$ almost surely. We denote by $\hat{z}_i = c_A \hat{G}^{1/2} z_i$ accordingly. After the transformation, we train the neural network via MFLD; that is, we optimize the objective $\mu \mapsto \frac{1}{n} \sum_{i=1}^{n} \ell(f_\mu(\hat{z}_i) y_i) + \lambda(\lambda_1 \mathbb{E}_\mu[\|X\|^2] + \mathrm{Ent}(\mu))$. Intuitively, this coordinate transform amplifies the informative coordinates ($j \in I_k$) and suppress the non-informative coordinates ($j \notin I_k$). Hence the covariance of the input features becomes more well-specified to the target signal $Y$ leading to a better LSI constant. We remark that such coordinate transformation is equivalent to employing an *anisotropic* weight decay regularization on the weight parameters $r(x) = \|x\|^2_{\hat{G}^{-1}}$.

Taken into account the sample complexity to estimate the gradient covariance, we obtain the following evaluation of the KL divergence between the prior distribution $\nu$ and a Bayes optimal solution $\mu^*$.

**Theorem 2.** *Assume that $d \max_{j' \in I_k^c} s_{j'}^2 = O(1)$. Suppose that $c_0$ is taken sufficiently small such that $\sum_{j=1}^{d} w_j^2 s_j^2 \leq 1$ almost surely for $w \sim \mathcal{U}(\mathcal{B}_{c_0})$ and $\mathbb{E}[|w_j|] = \Theta(1)$[2], and the regularization parameter $\hat{\lambda}_0$ is set to be $\hat{\lambda}_0 = \prod_{j' \in I_k} s_{j'}^2 \cdot \max_{j' \notin I_k} s_{j'}^2$. We assume that the sample size $n$ and the number of particles $N$ satisfies*

$$n \geq C_k \frac{k d \bar{R}^2 \log(2N/\delta)^2}{\prod_{j' \in I_k} s_{j'}^2}, \quad N \geq C_k \frac{k^2 d^2 \log(d/\delta)}{\max_{j' \notin I_k} s_{j'}^4}, \tag{11}$$

*for given $\delta \in (0,1)$, where $C_k$ is a constant depending on $k$. Then, for $\bar{R} = k$ and sufficiently small $C_k$, there exists $\mu^* \in \mathcal{P}$ such that $L(\mu^*) \leq \ell(0) - c_2$ and $\mathrm{KL}(\nu\|\mu^*) \leq R$ where*

$$R = c_1 k^2 \left( \frac{\max_{j' \in [d]} s_{j'}^2}{\min_{j' \in I_k} s_{j'}^2} + 1 \right) \log(k)^2,$$

*for a constant $c_1$ independent of the dimensionality $d$, with probability $1 - \delta$. Here, the probability is with respect to the randomness of training data and generating the initial parameters $(w_l)_{l=1}^{N}$.*

We make the following remarks on the theorem.

- This theorem implies a significant improvement on the LSI constant since $R$ is independent of $d$ as long as $\frac{\max_{j' \in [d]} s_{j'}^2}{\min_{j' \in I_k} s_{j'}^2} = \mathcal{O}(1)$, which is satisfied even for the isotropic setting. The dimension-free $R$ then implies that no exponential dependence is present in the computational complexity. Moreover, the runtime and the network width both "decouples" $k$ from the exponent in dimension dependence.

- In order to accurately estimate the gradient matrix, there is an additional cost in the statistical complexity. For the isotropic setting, (11) implies a sample complexity of $n = \Omega(d^{k+1})$, which matches the total sample size in the stochastic gradient descent procedure as in Barak et al. (2022).

- If the input is anisotropic so that $\prod_{j' \in I_k} s_{j'}^2 \gg d^{-k}$, then the sample complexity to estimate the informative direction is also improved. For instance, in the spiked setting (10), the sample complexity is improved to $n \asymp d^{(1-\alpha)k} d = d_{\mathrm{eff}}^k d$, and in the most extreme case, when the signal is well-specified by the principle components (i.e., denominator is $\Omega(1)$), the complexity becomes linear in $d$. This observation also demonstrates the benefit of structured data in feature learning.

**Tradeoff between statistical and computational complexity.** By comparing the complexity derived in Theorem 1 and Theorem 2, we observe a tradeoff between the statistical and computational complexity: estimating the gradient covariance matrix requires additional samples, but consequently the required width and iterations of MFLD significantly decrease. An interesting question is whether such tradeoff naturally occurs in more general data settings and feature learning procedures.

---

[2]This condition implicitly assumes $s_j^2 = O(1/d)$ ($\forall j$); we observe that $\sum_j w_j^2 s_j^2 \leq \max_j s_j^2 \sum_j w_j^2 = \max_j s_j^2 d$, which can be bounded by $O(1)$ only when $\max_j s_j^2 = O(1/d)$.

## 5    KERNEL LOWER BOUND FOR ANISOTROPIC PARITY

To complement our feature learning results, in this section we prove a classification lower bound for kernel methods on the (axis-aligned) anisotropic $k$-parity problem in Example 2 in a spiked covariance setting. We assume $y = y(z) = \text{sign}(\prod_{i=1}^{k} z_i)$, and for $k^* \leq k$, the input features satisfy

$$|z_i| = d^{\alpha/2} \text{ for } i = 1, \cdots, k^*. \qquad |z_i| = 1 \text{ for } i = k^*+1, \cdots, d. \tag{12}$$

This is to say, the first $k^*$ coordinates of the $k$-parity target function are aligned with the prominent directions of the input, but the target can also depend on an additional $k - k^*$ coordinates that are not amplified. It is clear that the following two settings are special cases of the above definition:

$(i)$ $\alpha = 0$: isotropic data.    $(ii)$ $k = k^*$: the (well-specified) spiked covariance model in (10).

We emphasize that most existing kernel lower bounds are tailored for regression with the squared error (Ghorbani et al., 2020; Hsu, 2021; Abbe et al., 2022). Even for the simplest isotropic setting $(i)$, to our knowledge the only classification lower bound for the parity problem is given in Wei et al. (2019) which only handles the $k = 2$ case (XOR).

Our lower bound applies to rotationally invariant kernels: $K(z, z') = h(\|z\|, \|z'\|, \langle z, z' \rangle)$ as considered in El Karoui (2010); Donhauser et al. (2021). In the hypercube setting, $\|z\|$ is fixed, and hence we may restrict ourselves to a positive semidefinite inner-product kernel which can be written as

$$K(z, z') = \sum_{l=0}^{\infty} \gamma_l \left( \frac{\langle z, z' \rangle}{d} \right)^l, \quad \{\gamma_l\}_{l=0}^{\infty} : \text{non-negative and bounded}.$$

Note that this covers the a wide range of NTK of fully-connected NNs (Liang et al., 2020; Donhauser et al., 2021). Given $n$ i.i.d. samples, we construct the kernel estimator $f_\beta(z)$ with $\beta \in \mathbb{R}^n$ chosen arbitrarily: $f_\beta(z) = \sum_{i=1}^{n} \beta_i K(z, z^{(i)})$. We have the following classification error lower bound.

**Theorem 3.** *Consider the spiked covariance setting in* (12)*. Given any fixed $\delta > 0$, if the sample size*

$$n \lesssim d^{\lfloor (1-\alpha)k^* \rfloor + k - k^* - \delta},$$

*then for sufficiently large $d$, with probability at least $0.99$ over the samples, for all choices of $\beta \in \mathbb{R}^n$, $f_\beta(z) = \sum_{i=1}^{n} \beta_i K(z, z^{(i)})$ will misclassify the sign of $y$ at $\Omega(1)$ fraction of the time, that is,*

$$\mathbb{P}_{z \sim P_Z} [f_\beta(z)y < 0] = \Omega(1).$$

The proof can be found in Appendix C. First, we lower bound the failure probability by the probability that $|f_\beta(z)|$ is large, by extending Wei et al. (2019) based on finer evaluation on the correlation $yK(z, z^{(i)})$. Then we reduce the problem to controlling the lowest eigenvalue of some kernel matrix.

We make the following remarks on the kernel lower bound.

- Recall that $\alpha \in (0, 1)$ controls the anisotropy of input features. When $\alpha \to 0$, the input becomes isotropic, and we obtain a $n \asymp d^{k-\delta}$ lower bound on the sample complexity for the classification error, which matches the regression lower bound in Ghorbani et al. (2019).

- In the "well-specified" setting where $k = k^*$ as in (10), the kernel sample complexity simplifies to $n \asymp d^{(1-\alpha)k} = d_{\text{eff}}^k$ which is strictly worse than that of MFLD given in Theorem 1 for $k > 1$. On the other hand, the required sample size is $d$ times better than the covariance estimation procedure in Theorem 2 (although we believe the factor $d$ stems from a technical artifact of the proof); however, in terms of computational complexity at test time, the kernel estimator needs to store $n$ training points which scales with $d^k$, whereas for the NN we only need to store $\text{poly}(d, k)$ neurons, which decouples the degree $k$ in the exponent of the dimension dependence.

## 6    EXPERIMENT

We validate our theoretical analysis by numerical experiments. We consider the anisotropic 3-sparse parity problem: $y = z_1 z_2 z_3$, $s_1 = s_2 = s_3 = \kappa/\sqrt{d}$, and $s_4 = \cdots = 1/\sqrt{d}$. We fixed $d = 300$ and varied $n$ and $\kappa$ to train the neural network (2) with the logistic loss. More details are in Appendix D.

In Figure 1 we plot the test accuracy as a function of the sample size $n$ and $\kappa$, which controls the level of anisotropy. As clearly seen, increasing $\kappa$ enables smaller the model to learn the problem with smaller sample complexity $n$, which demonstrates how anisotropy helps learning. Moreover, let us focus on the "phase transition" boundary between yellow and blue regions. According to Theorem 1, the classification error is bounded by $\sum_{j \in I_k} s_j^{-2}/n = \kappa^{-2}d/n$ up to a constant, which predicts that there would be a boundary around $\kappa^2 = \Theta(n)$, as indicated by the red line in the figure. We therefore conclude that the empirical findings match the theoretical result in Theorem 1.

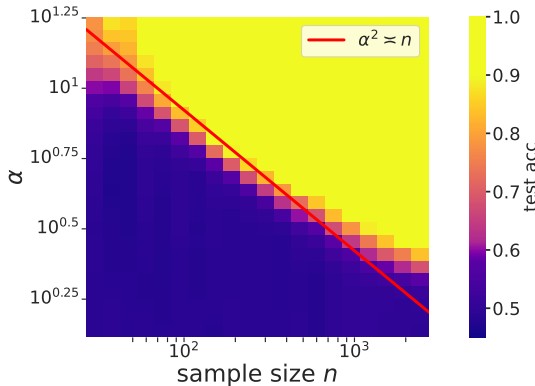

Figure 1: Test accuracy of NN trained by MFLD to learn an anisotropic $d$-dimensional 3-parity problem.

## 7 CONCLUSION

We studied the interplay between structured data (in the form of input anisotropy) and the efficiency of feature learning in the context of classifying $k$-sparse parity using two-layer neural network optimized by noisy gradient descent (mean-field Langevin dynamics). We showed that anisotropy can improve both the statistical and computational complexity of MFLD, leading to a separation from kernel methods (including neural networks in the NTK regime). Interesting future directions include $(i)$ extending this observation to a more general class of target functions, $(ii)$ improving the sample complexity to obtain the covariance estimator in Theorem 2, and $(iii)$ deriving a more precise description on the tradeoff between statistical and computational complexity in NN training.

## ACKNOWLEDGEMENTS

TS was partially supported by JSPS KAKENHI (20H00576) and JST CREST (JPMJCR2115, JP-MJCR2015). KO was partially supported by JST, ACT-X Grant Number JPMJAX23C4, JAPAN. AN was partially supported by JSPS KAKENHI (22H03650).

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

——————————— **Appendix** ———————————

## A  PROOFS OF PROPOSITIONS 3 AND 4 AND COROLLARY 1

*Proof of Proposition 3.* We follow the proof strategy from Suzuki et al. (2023b). Remember that

$$h_x(z) = \bar{R}[\tanh(z^\top x_1 + x_2) + 2\tanh(x_3)]/3.$$

Let $b_i = 2i - k$ for $i = 0, \dots, k$, let $\zeta > 0$ be the positive real such that $\mathbb{E}_{u \sim N(0,1)}[2\tanh(\zeta + u)] = 1$ (note that, this also yields $\mathbb{E}_{u \sim N(0,1)}[2\tanh(-\zeta + u)] = -1$ by the symmetric property of $\tanh$ and the Gaussian distribution). Let

$$\Sigma := \begin{pmatrix} I/(2\lambda_1) & 0 & 0 \\ 0 & 1/(2\lambda_1) & 0 \\ 0 & 0 & 1 \end{pmatrix} \in \mathbb{R}^{(d+1+1) \times (d+1+1)},$$

and $\rho > 1$ be a constant which will be adjusted later on. Then, for $\xi_{2j} := [\log(\rho k)\phi^\top, -\log(\rho k)(b_j - 1), \zeta]^\top \in \mathbb{R}^{\bar{d}}$ and $\xi_{2j+1} := -[\log(\rho k)\phi^\top, -\log(\rho k)(b_j + 1), \zeta]^\top \in \mathbb{R}^{\bar{d}}$ for $j = 0, \dots, k$, we define

$$\hat{\mu}_{2j} := N(\xi_{2j}, \Sigma), \quad \hat{\mu}_{2j+1} := N(\xi_{2j+1}, \Sigma).$$

By noticing that for $z \in \text{supp}(P_Z)$ there exists $\tilde{z} \in \{\pm 1/\sqrt{d}\}^d$ such that $z = A\tilde{z}$, we can see that

$$\mathbb{E}_{x \sim \hat{\mu}_{2j}}[h_x(z)] = \bar{R}\mathbb{E}_{u \sim N(0, 1/\lambda_1)}\{\tanh[\log(\rho k)(\langle \tilde{\phi}, \tilde{z} \rangle - (b_j - 1)) + u] + 1\}/3$$

because we have

$$\langle x_1, z \rangle + x_2 = \log(\rho k)(\langle \phi, z \rangle - (b_j - 1)) + \sum_{i=1}^d u_i z_i + u_{d+1}$$

$$= \log(\rho k)(\langle A^{-1}\tilde{\phi}, A\tilde{z} \rangle - (b_j - 1)) + \sum_{i=1}^d u_i z_i + u_{d+1},$$

for $x \sim N([\phi^\top, (b_j - 1)]^\top, I/(2\lambda_1))$ where $u_i \sim N(0, 1/(2\lambda_1))$ (i.i.d.) and $\sum_{i=1}^d u_i z_i + u_{d+1}$ obeys the Gaussian distribution with mean 0 and variance $\frac{1}{2\lambda_1}\|z\|^2 + \frac{1}{2\lambda_1} = \frac{1}{2\lambda_1}(1 + \|z\|^2) \le \frac{1}{\lambda_1}$ for all $z \in \text{supp}(P_Z)$, where we used the assumption on $A$. In the same vein, we also have

$$\mathbb{E}_{x \sim \hat{\mu}_{2j+1}}[h_x(z)] = -\bar{R}\mathbb{E}_{u \sim N(0, (1+\|z\|^2)/2\lambda_1)}\{\tanh[\log(\rho k)(\langle \tilde{\phi}, \tilde{z} \rangle - (b_j + 1)) + u] + 1\}/3.$$

Here, define $|\tilde{z}|_0 := |\{i \in I_k \mid \tilde{z}_i > 0\}|$ for $\tilde{z} \in \text{supp}(P_{\tilde{Z}})$ which is the number of positive elements of $z$ in the informative index set $I_k$. For a fixed number $j \in \{0, \dots, k\}$, we let

$$f_1(z; u) = \{\tanh[\log(\rho k)(\langle \tilde{\phi}, \tilde{z} \rangle - (b_j - 1)) + u] + 1\}/3,$$
$$f_2(z; u) = \{\tanh[\log(\rho k)(\langle \tilde{\phi}, \tilde{z} \rangle - (b_j + 1)) + u] + 1\}/3,$$

then we can see that

$$f_1(z; 0) = \begin{cases} O(1/(\rho k)) & (|\tilde{z}|_0 < j), \\ 1 - O(1/(\rho k)) & (|\tilde{z}|_0 \ge j), \end{cases}$$

and

$$f_2(z; 0) = \begin{cases} O(1/(\rho k)) & (|\tilde{z}|_0 < j + 1), \\ 1 - O(1/(\rho k)) & (|\tilde{z}|_0 \ge j + 1), \end{cases}$$

because $\langle \tilde{\phi}, \tilde{z} \rangle - b_j = \sum_{j'=1}^k \text{sign}(\tilde{z}_{j'}) - b_j = 2|\tilde{z}|_0 - k - b_j = 2(|\tilde{z}|_0 - j)$. Hence, we have that

$$f(z; u) := f_1(z; u) - f_2(z; u) = \begin{cases} \Omega(1) & (|\tilde{z}|_0 = j), \\ O(1/(\rho k)) & (\text{otherwise}), \end{cases}$$

and $f(z; u) > 0$ for $|\tilde{z}|_0 = j$. Then, since $\tanh(u) + 1 = \frac{e^u - e^{-u}}{e^u + e^{-u}} + 1 = \frac{2}{1 + e^{-2u}}$, if $|\tilde{z}|_0 = j$ and $|u| \le 1/\lambda_1$,

$$f(z; u) \ge \Omega(1),$$

and if $|\tilde{z}|_0 \neq j$ and $|u| \leq \log(\rho k)/2$,

$$f(z; u) \leq O(1/(\rho k)).$$

Therefore, when $|\tilde{z}|_0 = j$,

$$\mathbb{E}_{u \sim N(0,(1+\|z\|^2)/2\lambda_1)}[f(z; u)] \geq \int_{-1/\lambda_1}^{1/\lambda_1} f(z; u)g(u)\mathrm{d}u > \Omega(1).$$

where $g$ is the density function of $N(0, (1 + \|z\|^2)/2\lambda_1)$, and when $|\tilde{z}|_0 \neq j$,

$$\mathbb{E}_{u \sim N(0,(1+\|z\|^2)/2\lambda_1)}[f(z; u)] \leq \int_{-\log(\rho k)/2}^{\log(\rho k)/2} f(z; u)g(u)\mathrm{d}u + \int_{|u| \geq \log(\rho k)/2} f(z; u)g(u)\mathrm{d}z$$
$$\leq O(1/(\rho k)) + O\left(\frac{\exp(-\lambda_1 \log(\rho k)^2/2)}{\log(\rho k)}\right)$$
$$= O(1/(\rho k)),$$

where we used the upper-tail inequality of the Gaussian distribution in the second inequality. Hence, it holds that

$$\hat{f}_i(z) := \mathbb{E}_{x \sim \hat{\mu}_{2i}}[h_x(z)] + \mathbb{E}_{x \sim \hat{\mu}_{2i+1}}[h_x(z)] = \begin{cases} \Omega(k) & (|\tilde{z}|_0 = j), \\ O(1/\rho) & (\text{otherwise}), \end{cases}$$

because $\bar{R} = k$. Therefore, by taking $\rho > 1$ sufficiently large, we also have

$$\hat{f}(z) := \frac{1}{2(k+1)} \sum_{i=0}^{k} (-1)^i \hat{f}_i(z) = \begin{cases} \Omega(1) & (|\tilde{z}|_0 \text{ is even}), \\ -\Omega(1) & (|\tilde{z}|_0 \text{ is odd}), \end{cases}$$

where the constant hidden in $\Omega(\cdot)$ is uniform over any $|\tilde{z}|_0$. Hence, there exists $c_2' > 0$ such that $Y\hat{f}(Z) > c_2'$ almost surely. Now if we let $\mu_{\langle a \rangle}(B) := \mu(aB)$ for $a \in \mathbb{R}$, a probability measure $\mu$ and a measurable set $B$, we can see that $\hat{f}$ is represented as

$$\hat{f}(\cdot) = \mathbb{E}_{x \sim \mu^*}[h_x(\cdot)],$$

where

$$\mu^* = \frac{1}{2(k+1)} \sum_{i=0}^{k} (\hat{\mu}_{2i,\langle (-1)^i \rangle} + \hat{\mu}_{2i+1,\langle (-1)^i \rangle}).$$

Then, by letting $c_2 = \ell(0) - \ell(c_2')$, we have

$$L(\mu^*) \leq \ell(0) - c_2.$$

Next, we bound the KL divergence between $\nu$ and $\mu^*$. The convexity of KL divergence yields that

$$\mathrm{KL}(\nu, \mu^*) \leq \frac{1}{2(k+1)} \sum_{i=0}^{k} (\mathrm{KL}(\nu, \hat{\mu}_{2i}) + \mathrm{KL}(\nu, \hat{\mu}_{2i+1}))$$
$$\leq \lambda_1 \log(\rho k)^2 [\|\phi\|^2 + (\max_j |b_j| + 1)^2] + \log(1/(2\lambda_1)) + \lambda_1(1 + \zeta^2)$$
$$= O\left(\log(k)^2 \left(\|\phi\|^2 + k^2\right)\right).$$

This gives the assertion. $\qquad \square$

Next, we prove Proposition 4.

*Proof of Proposition 4.* The proof of this statement resembles Proposition 4 of Suzuki et al. (2023b). The key step in their proof is to show that the optimal solution satisfies

$$|f_{\mu[\lambda]}(z)| = |f_{\mu[\lambda]}(z')|$$

for any $z, z' \in \mathrm{supp}(P_Z)$. We prove that this still holds in our general setting. Let $T_A : \mathbb{R}^{\bar{d}} \to \mathbb{R}$ be

$$T_A x = (Ax_1, x_2, x_3),$$

where $x = (x_1, x_2, x_3)$ for $x_1 \in \mathbb{R}^d$, $x_2 \in \mathbb{R}$ and $x_3 \in \mathbb{R}$. Then, we can see that

$$f_\mu(z) = f_{T_{A\#}\mu}(\tilde{z})$$

for $\mu \in \mathcal{P}$ and $T_{A\#}$ is the push-forward with respect to $T_A$, and $z = A\tilde{z}$. Based on this coordinate transform, we can reduce the problem to the standard parity setting where the input obeys the uniform distribution on $\{\pm 1/\sqrt{d}\}^d$. According to this coordinate transform, the prior distribution $\nu$ is transformed to $\nu_A := T_{A\#}\nu$, which is again a normal distribution with mean 0 and variance $AA^\top/(2\lambda_1)$. We also let $T_j$ be the map which flips the sign of the $i$-th coordinate. Then, the key argument in the proof of Suzuki et al. (2023b) is to show that

$$\mathrm{KL}(\nu_A||\mu) = K(\nu_A||T_{j\#}\mu)$$

for a measure $\mu \in \mathcal{P}$ (which is supposed to be $T_{A\#}\hat{\mu}$ for a population risk minimizer $\hat{\mu}$). This equality is true because the normal distribution is point symmetric. Indeed, we have

$$\mathrm{KL}(\nu_A||\mu) = \mathrm{KL}(T_{j\#}\nu_A||T_{j\#}\mu) = \mathrm{KL}(\nu_A||T_{j\#}\mu),$$

where the first equality is by the invariance of the KL divergence against any bijective coordinate transform and the second equality is by the point symmetry of the normal distribution. Then, following the same argument to Suzuki et al. (2023b), we obtain the assertion. □

Finally, Proposition 1 can be obtained as a corollary of Proposition 3 where we set $A = \mathrm{diag}\left(s_1\sqrt{d}, s_2\sqrt{d}, \dots, s_d\sqrt{d}\right)$. For this setting, we can easily see that

$$\|\phi\|^2 = \sum_{j \in I_k} s_j^{-2}.$$

Combining with this evaluation and the fact that

$$k = \sum_{i \in I_k} 1 = \sum_{i \in I_k} s_i s_i^{-1} \le \sqrt{\sum_{i \in I_k} s_i^2}\sqrt{\sum_{i \in I_k} s_i^{-2}} \le \sqrt{\sum_{i \in I_k} s_i^{-2}}$$

we obtain the assertion.

## B  ESTIMATING THE INFORMATION MATRIX

Without loss of generality, we may take $I_k = \{1, \dots, k\}$. Let $\sigma(w^\top z) = h_x(z)$ for $(x_1, x_2, x_3) = (w, b_1, b_2)$ for a fixed $b_1$ and $b_2$. Then,

$$\sigma(w^\top z) = \sum_{\ell=0}^{\infty} \frac{1}{\ell!} \underbrace{\sigma^{(\ell)}(0)}_{=:c_\ell}(w^\top z)^\ell.$$

Note that the gradient of the loss with respect to $w_j$ can be written as

$$g_j(w) = \frac{1}{n}\sum_{i=1}^{n} \ell'(y_i f_{\mu_0}(z_i))y_i z_{i,j}\sigma'(w^\top z_i).$$

Suppose that $f_{\mu_0}(z_i) = 0$, then since $Y = \prod_{j \in I_k}(s_j^{-1}Z_j)$, its expectation can be expressed as

$$\bar{g}_j(w) := \mathbb{E}\left[\prod_{j' \in I_k}(s_{j'}^{-1}Z_{j'})Z_j\sigma'(w^\top Z)\right].$$

(1) If $j \in I_k$, then we have that

$$\bar{g}_j(w) := s_j\prod_{j' \in I_k \setminus j} s_{j'}^{-1}\mathbb{E}\left[\prod_{j' \in I_k \setminus j} Z_{j'}\sigma'(w^\top Z)\right].$$

Then, by the Taylor expansion of $\sigma$, it holds that

$$
\begin{aligned}
\bar{g}_j(w) =& s_j \prod_{j' \in I_k \setminus j} s_{j'}^{-1} \left( \sum_{\ell=0}^{k-1} \frac{1}{\ell!} \partial_{\tilde{\theta}}^{(\ell)} \mathbb{E} \left[ \prod_{j' \in I_k \setminus j} Z_{j'} \sigma'((\tilde{\theta} w)^\top Z) \right] \Bigg|_{\tilde{\theta}=0} \right. \\
& \left. + \sum_{\ell=k}^{\infty} \frac{1}{\ell!} \partial_{\tilde{\theta}}^{(\ell)} \mathbb{E} \left[ \prod_{j' \in I_k \setminus j} Z_{j'} \sigma'((\tilde{\theta} w)^\top Z) \right] \Bigg|_{\tilde{\theta}=0} \right) \\
=& s_j \prod_{j' \in I_k \setminus j} s_{j'}^{-1} \left( \mathbb{E} \left[ \prod_{j' \in I_k \setminus j} Z_{j'} \frac{c_k}{(k-1)!} (w^\top Z)^{k-1} \right] \right. \\
& \left. + \sum_{\ell=k}^{\infty} \mathbb{E} \left[ \prod_{j' \in I_k \setminus j} Z_{j'} \cdot \frac{c_{\ell+1}}{\ell!} (w^\top Z)^\ell \right] \right) \\
=& s_j \prod_{j' \in I_k \setminus j} s_{j'}^{-1} \left( \prod_{j' \in I_k \setminus j} s_{j'}^2 \frac{c_k}{(k-1)!} (k-1)! \prod_{j' \in I_k \setminus j} w_{j'} + \underbrace{(\text{higher order term})}_{=:(a)} \right) \\
=& c_k \cdot \prod_{j' \in I_k} s_{j'} \cdot \prod_{j' \in I_k \setminus j} w_{j'} + (\text{higher order term}).
\end{aligned}
$$

The higher order term $(a)$ in the above expression can be evaluated as

$$
\begin{aligned}
& \sum_{\ell=k}^{\infty} \mathbb{E} \left[ \prod_{j' \in I_k \setminus j} Z_{j'} \cdot \frac{c_{\ell+1}}{\ell!} (w^\top Z)^\ell \right] \\
=& \sum_{\ell=k}^{\infty} \mathbb{E} \left\{ \prod_{j' \in I_k \setminus j} Z_{j'} \cdot \frac{c_{\ell+1}}{\ell!} \left[ \sum_{\{j_1,\dots,j_{k-1}\}=I_k \setminus j} \sum_{0 \le \ell_1 < \ell_2 < \cdots < \ell_{k-1} < \ell} \left( \sum_{j' \in (I_k \setminus j)^c} w_{j'} Z_{j'} \right)^{\ell_1} w_{j_1} Z_{j_1} \right. \right. \\
& \left. \cdot \left( \sum_{j' \in (I_k \setminus \{j,j_1\})^c} w_{j'} Z_{j'} \right)^{\ell_2-\ell_1-1} w_{j_2} Z_{j_2} \cdot \left( \sum_{j' \in (I_k \setminus \{j,j_1,j_2\})^c} w_{j'} Z_{j'} \right)^{\ell_3-\ell_2-1} w_{j_3} Z_{j_3} \right. \\
& \left. \left. \cdots \left( \sum_{j' \in (I_k \setminus \{j,j_1,j_2,\dots,j_{k-2}\})^c} w_{j'} Z_{j'} \right)^{\ell_{k-1}-\ell_{k-2}-1} w_{j_{k-1}} Z_{j_{k-1}} \cdot (w^\top Z)^{\ell-\ell_{k-1}-1} \right] \right\} \\
=& \sum_{\ell=k}^{\infty} \mathbb{E} \left\{ \prod_{j' \in I_k \setminus j} (s_{j'}^2 w_{j'}) \cdot \frac{c_{\ell+1}}{\ell!} \left[ \sum_{\{j_1,\dots,j_{k-1}\}=I_k \setminus j} \sum_{0 \le \ell_1 < \ell_2 < \cdots < \ell_{k-1} < \ell} \left( \sum_{j' \in (I_k \setminus j)^c} w_{j'} Z_{j'} \right)^{\ell_1} \right. \right. \\
& \cdot \left( \sum_{j' \in (I_k \setminus \{j,j_1\})^c} w_{j'} Z_{j'} \right)^{\ell_2-\ell_1-1} \cdot \left( \sum_{j' \in (I_k \setminus \{j,j_1,j_2\})^c} w_{j'} Z_{j'} \right)^{\ell_3-\ell_2-1} \cdots \\
& \left. \left. \left( \sum_{j' \in (I_k \setminus \{j,j_1,j_2,\dots,j_{k-2}\})^c} w_{j'} Z_{j'} \right)^{\ell_{k-1}-\ell_{k-2}-1} \cdot (w^\top Z)^{\ell-\ell_{k-1}-1} \right] \right\}.
\end{aligned}
$$

Then, applying Hölder's inequality, we can bound the right hand side as

$$
\sum_{\ell=k}^{\infty} \prod_{j' \in I_k \setminus j} (s_{j'}^2 w_{j'}) \cdot \frac{c_{\ell+1}}{\ell!} \cdot \sum_{\{j_1,\dots,j_{k-1}\}=I_k \setminus j} \sum_{0 \le \ell_1 < \ell_2 < \cdots < \ell_{k-1} < \ell} \mathbb{E} \left[ \left| \sum_{j' \in (I_k \setminus j)^c} w_{j'} Z_{j'} \right|^{\ell-k+1} \right]^{\frac{\ell_1}{\ell-k+1}}
$$

$$\cdot \mathbb{E}\left[\left|\sum_{j'\in(I_k\setminus\{j,j_1\})^c} w_{j'}Z_{j'}\right|^{\ell-k+1}\right]^{\frac{\ell_2-\ell_1+1}{\ell-k+1}} \cdots \mathbb{E}\left[\left|\sum_{j'\in(I_k\setminus\{j,j_1,\ldots,j_{k-2}\})^c} w_{j'}Z_{j'}\right|^{\ell-k+1}\right]^{\frac{\ell_{k-1}-\ell_{k-2}+1}{\ell-k+1}}$$

$$\cdot \mathbb{E}\left[\left|w^\top Z\right|^{\ell-k+1}\right]^{\frac{\ell-\ell_{k-1}+1}{\ell-k+1}}.$$

Now, we use the moment bound of sub-Gaussian random variables to bound each term $\mathbb{E}[|\sum_{j'\in(I_k\setminus\{j,j_1,\ldots,j_a\})^c} w_{j'}Z_{j'}|^{\ell-k+1}]^{\frac{\ell_a-\ell_{a-1}+1}{\ell-k+1}}$. Indeed, we can see that, for any index set $J \subset I$, $w_J^\top Z_J$ is a sub-Gaussian random variable with parameter $\|w_J \odot s_J\|$[3] where a sub-Gaussian random variable $X$ with a parameter $u$ satisfied $\mathbb{E}[|X|^\ell] \leq (cu)^\ell \ell^{\ell/2}$ ($\forall \ell \geq 1$) with an absolute constant $c$ (see Proposition 2.5.2 of Vershynin (2020), for example). Hence, by noticing that $\|w_J \odot s_J\|^2 \leq \|w \odot s\|^2$ and $\ell_1 + \sum_{b=2}^{k-1}(\ell_b - \ell_{b-1} + 1) + \ell - \ell_{k-1} + 1 = \ell - k + 1$, the right hand side can be bounded by

$$\prod_{j'\in I_k\setminus j} s_{j'}^2 \cdot \prod_{j'\in I_k\setminus j} w_{j'} \cdot \sum_{\ell=k}^\infty \frac{c_{\ell+1}}{\ell!}(k-1)!\frac{\ell!}{(\ell-k+1)!(k-1)!}(c\|w\odot s\|)^{\ell-k+1}(\ell-k+1)^{(\ell-k+1)/2}$$

$$= \prod_{j'\in I_k\setminus j} s_{j'}^2 \cdot \prod_{j'\in I_k\setminus j} w_{j'} \cdot \sum_{\ell=k}^\infty c_{\ell+1}(c\|w\odot s\|)^{\ell-k+1}\frac{(\ell-k+1)^{(\ell-k+1)/2}}{(\ell-k+1)!}.$$

Then, by the Stirling's formula, the right hand side can be bounded by

$$\prod_{j'\in I_k\setminus j} s_{j'}^2 \cdot \prod_{j'\in I_k\setminus j} |w_{j'}| \cdot \sum_{\ell=k}^\infty c_{\ell+1}\|w\odot s\|^{\ell-k+1}\frac{(\ell-k+1)^{(\ell-k+1)/2}}{\sqrt{2\pi}(\ell-k+1)^{\ell-k+1+1/2}e^{-(\ell-k+1)}}$$

$$= \prod_{j'\in I_k\setminus j} s_{j'}^2 \cdot \prod_{j'\in I_k\setminus j} |w_{j'}| \cdot \sum_{\ell=k}^\infty c_{\ell+1}\|w\odot s\|^{\ell-k+1}\frac{1}{\sqrt{2\pi}}\left(\frac{e}{(\ell-k+1)^{1/2}}\right)^{\ell-k+1}\frac{1}{(\ell-k+1)^{1/2}}$$

$$\leq \frac{c_k}{2}\prod_{j'\in I_k\setminus j} s_{j'}^2 \cdot \prod_{j'\in I_k\setminus j} |w_{j'}|,$$

where we used the assumption $\|w \odot s\|$ is sufficiently small such that $\sum_{\ell=k}^\infty c_{\ell+1}(c\|w\odot s\|)^{\ell-k+1}\frac{1}{\sqrt{2\pi}}\left(\frac{e}{(\ell-k+1)^{1/2}}\right)^{\ell-k+1}\frac{1}{(\ell-k+1)^{1/2}} \leq \frac{c_k}{2}$. Therefore, we can see that

$$\bar{g}_j(w) = c_k \cdot \prod_{j'\in I_k} s_{j'} \cdot \prod_{j'\in I_k\setminus j} w_{j'} + \text{(higher order term)},$$

$$|\bar{g}_j(w)| \geq \frac{c_k}{2} \cdot \prod_{j'\in I_k} s_{j'} \cdot \prod_{j'\in I_k\setminus j} |w_{j'}|,$$

$$|\bar{g}_j(w)| \leq \frac{3}{2}c_k \cdot \prod_{j'\in I_k} s_{j'} \cdot \prod_{j'\in I_k\setminus j} |w_{j'}|. \tag{13}$$

(2) In the same vein, we also have that, for $j \notin I_k$, it holds that

$$\bar{g}_j(w) = c_{k+2} \cdot \prod_{j'\in I_k\cup j} s_{j'} \cdot \prod_{j'\in I_k\cup j} w_{j'} + \text{(higher order term)},$$

$$|\bar{g}_j(w)| \leq 2c_{k+2} \cdot \prod_{j'\in I_k\cup j} s_{j'} \cdot \prod_{j'\in I_k\cup j} |w_{j'}|. \tag{14}$$

Next, we show the concentration of the empirical gradient $g_j(w)$ around its expectation. Observe that

$$\sup_{Y,Z}|\ell'(Yf_{\mu_0}(Z))YZ_j\sigma'(w^\top Z)| \leq \bar{R}s_j,$$

---

[3]Here, $x \odot y := (x_j y_j)_{j=1}^d$.

$$\mathrm{Var}_{Y,Z}[\ell'(Yf_{\mu_0}(Z))YZ_j\sigma'(w^\top Z)] \leq \bar{R}^2 s_j^2.$$

Therefore, by Bernstein's inequality, we obtain that

$$P\left(|g_j(w) - \bar{g}_j(w)| \geq \frac{4\bar{R}s_j}{\sqrt{n}}\log(2/\delta)\right) \leq \delta$$

for any $\delta \in (0,1)$. Hence, if we let $n$

$$n \geq \frac{16k\bar{R}^2\log(2N/\delta)^2 d}{\left(C_0 c_k \cdot \prod_{j'\in I_k} s_{j'}\right)^2},$$

for a sufficiently small constant $C_0$, then we have that

$$|g_j(w_l) - \bar{g}_j(w_l)| \leq C_0 c_k \prod_{j'\in I_k} s_{j'} \cdot s_j / \sqrt{kd}, \tag{15}$$

uniformly over $l = 1, \ldots, N$ with probability $\delta$.

For that purpose, we evaluate the expectations of $g_{j_1}(w)g_{j_2}(w)$ carefully. Let $H(w) = \sum_{\ell=k}^{\infty} \frac{c_{\ell+1}}{(\ell-k+1)!}\mathbb{E}_Z\left[(w^\top Z)^{\ell-k+1}\right] = \frac{1}{2}\|w \odot s\|^2 + \sum_{\ell=0}^{\infty} \frac{c_{k+4+2\ell}}{(4+2\ell)!}\mathbb{E}_Z\left[(w^\top Z)^{4+2\ell}\right]$. We evaluate for each condition on $j_1$ and $j_2$.

(a) If $j_1 = j_2 \in I_k$, then it holds that

$$\mathbb{E}_W[\bar{g}_{j_1}(W)\bar{g}_{j_1}(W)] = c_k^2 \prod_{j'\in I_k} s_{j'}^2 \mathbb{E}_W\left[\prod_{j'\in I_k \setminus j_1} W_{j'}^2(1 + H(W))^2\right]$$

$$= \Omega\left(\prod_{j'\in I_k} s_{j'}^2\right).$$

(b) If $j_1 \neq j_2$ and $j_1, j_2 \in I_k$, then it holds that

$$\mathbb{E}_W[\bar{g}_{j_1}(W)\bar{g}_{j_2}(W)] = c_k^2 \prod_{j'\in I_k} s_{j'}^2 \mathbb{E}\left[\prod_{j'\in I_k \setminus \{j_1, j_2\}} W_{j'}^2 \cdot W_{j_1}W_{j_2}(1 + H(W))^2\right] = 0,$$

where we used that the distribution of $W$ is symmetric and $H(W)$ satisfies $H(W) = H(-W)$.

(c) If $j_1 \neq j_2$ and $j_1 \in I_k$ and $j_2 \notin I_k$, then

$$\mathbb{E}_W[\bar{g}_{j_1}(W)\bar{g}_{j_2}(W)] = c_k c_{k+2} \prod_{j'\in I_k} s_{j'}^2 s_{j_2} \mathbb{E}\left[\prod_{j'\in I_k \setminus j_1} W_{j'}^2 \cdot W_{j_2}(1 + H(W))^2\right] = 0.$$

(d) If $j_1 \notin I_k$ and $j_2 \notin I_k$, then

$$\mathbb{E}_W[\bar{g}_{j_1}(W)\bar{g}_{j_2}(W)] = c_{k+2}^2 \prod_{j'\in I_k} s_{j'}^2 s_{j_1} s_{j_2} \mathbb{E}\left[\prod_{j'\in I_k} W_{j'}^2 \cdot W_{j_1}W_{j_2}(1 + H(W))^2\right]$$

$$= \begin{cases} 0 & (j_1 \neq j_2), \\ \mathcal{O}(\prod_{j'\in I_k \cup j_1} s_{j'}^2) & (j_1 = j_2). \end{cases}$$

Summarizing these evaluations, we can see that $\bar{G} = (\bar{G}_{j_1,j_2})_{j_1=1,j_2=1}^{d,d} \in \mathbb{R}^{d \times d}$ defined by

$$\bar{G}_{j_1,j_2} = \mathbb{E}_W[\bar{g}_{j_1}(W)\bar{g}_{j_2}(W)]$$

is a diagonal matrix where $\bar{G}_{j_1,j_1}$ for $j_1 \in I_k$ has larger values than that for $j_1 \notin I_k$. We define its empirical average version $G = (G_{j_1,j_2})_{j_1=1,j_2=1}^{d,d} \in \mathbb{R}^{d \times d}$ as

$$G_{j_1,j_2} = \frac{1}{N}\sum_{l=1}^{N} g_{j_1}(w_l)g_{j_2}(w_l).$$

Now, we show the concentration of $G$ around its population version $\bar{G}$. Note that

$$\frac{1}{N}\sum_{l=1}^{N} g_{j_1}(w_l)g_{j_2}(w_l) = \frac{1}{N}\sum_{l=1}^{N}(g_{j_1}(w_l) - \bar{g}_{j_1}(w_l) + \bar{g}_{j_1}(w_l))(g_{j_2}(w_l) - \bar{g}_{j_2}(w_l) + \bar{g}_{j_2}(w_l))$$

$$= \frac{1}{N}\sum_{l=1}^{N}(g_{j_1}(w_l) - \bar{g}_{j_1}(w_l))(g_{j_2}(w_l) - \bar{g}_{j_2}(w_l))$$

$$+ \frac{1}{N}\sum_{l=1}^{N}(g_{j_1}(w_l) - \bar{g}_{j_1}(w_l))\bar{g}_{j_2}(w_l)$$

$$+ \frac{1}{N}\sum_{l=1}^{N}(g_{j_2}(w_l) - \bar{g}_{j_2}(w_l))\bar{g}_{j_1}(w_l)$$

$$+ \frac{1}{N}\sum_{l=1}^{N}\bar{g}_{j_1}(w_l)\bar{g}_{j_2}(w_l).$$

Therefore, if we let $\Delta G_{j_1,j_2} = G_{j_1,j_2} - \bar{G}_{j_1,j_2}$, $\hat{G}_{j_1,j_2} = \frac{1}{N}\sum_{l=1}^{N}\bar{g}_{j_1}(w_l)\bar{g}_{j_2}(w_l)$ and $\dot{G}_{j_1,j_2} := \frac{1}{N}\sum_{l=1}^{N}(g_{j_1}(w_l) - \bar{g}_{j_1}(w_l))(g_{j_2}(w_l) - \bar{g}_{j_2}(w_l))$, then for any $u \in \mathbb{R}^d$, it holds that

$$|u^\top \Delta G u| \leq 2u^\top \left(\frac{1}{N}\sum_{l=1}^{N}(g(w_l) - \bar{g}(w_l))(g(w_l) - \bar{g}(w_l))^\top\right)u + 2u^\top\left(\frac{1}{N}\sum_{l=1}^{N}\bar{g}(w_l)\bar{g}(w_l)^\top\right)u$$

$$\leq 2u^\top \dot{G}u + 2u^\top(\hat{G} - \bar{G})u + 2u^\top \bar{G}u. \tag{16}$$

Here, the term $\hat{G} - \bar{G}$ can be bounded by the matrix Bernstein's inequality as

$$P\left[\|\hat{G} - \bar{G}\|_{\mathrm{op}} \geq K\left(\sqrt{\frac{Q^2(t + \log(d))}{N}} + \frac{(t + \log(d))Q}{N}\right)\right] \leq \exp(-t),$$

where $K$ is an absolute constant and $Q = d\prod_{j' \in I_k} s_{j'}^2$ because $\|\bar{g}(w_l)\bar{g}^\top(w_l)\|_{\mathrm{op}} \leq O(Q)$. Therefore, taking $N = \Omega(d^2 k^2 \log(d/\delta)/(C_0 \max_{j' \notin I_k} s_{j'}^4))$ for sufficiently small $C_0$ yields that

$$\|\hat{G} - \bar{G}\|_{\mathrm{op}} = \mathcal{O}\left(C_0 \prod_{j' \in I_k} s_{j'}^2 \cdot \max_{j' \notin I_k} s_{j'}^2/k\right),$$

with probability $1 - \delta$. In the same manner, we also have that

$$G \succeq \frac{1}{2}\bar{G} + (\hat{G} - \bar{G}) - \dot{G}. \tag{17}$$

In the following, we let $Q_1 := \prod_{j' \in I_k} s_{j'}^2$, and $Q_2 := \prod_{j' \in I_k} s_{j'}^2 \cdot \max_{j' \notin I_k} s_{j'}^2$.

Then, by modifying the objective as

$$L(\mu) + \lambda_1 \mathbb{E}_\mu[\|X\|^2_{(G + \hat{\lambda}_0 I)^{-1}}]$$

with a regularization parameter $\hat{\lambda}_0 = Q_2$. This is equivalent to the alternative objective $L(\mu) + \lambda_1 \mathbb{E}_\mu[\|X\|^2]$ where the input is transformed as $Z \leftarrow A\tilde{Z}$ where $A = c_A\sqrt{G + \hat{\lambda}_0 I}B$ with $B = \mathrm{diag}\left(s_1\sqrt{d}, \ldots, s_d\sqrt{d}\right)$ and a constant $c_A = \mathcal{O}((kQ_1 \max_{j'} s_{j'}^2)^{-1/2})$ such that $\|A\tilde{Z}\| \leq 1$. Indeed, we can take such $c_A$ because

$$\left\|\sqrt{G + \hat{\lambda}_0 I}B\tilde{Z}\right\|^2 \leq \tilde{Z}^\top B\left(2\dot{G} + (\hat{G} - \bar{G}) + 3\bar{G}\right)B\tilde{Z}$$

$$\lesssim \left(\sum_{j_1,j_2=1}^{d} s_{j_1}^2 s_{j_2}^2\right)Q_1/(kd) + Q_2(d\max_{j'} s_{j'}^2) + \max_{j'} s_{j'}^2(kQ_1 + (d-k)Q_2)$$

$$\lesssim kQ_1 \max_{j'} s_{j'}^2,$$

where we used the assumption of $\sum_{j=1}^d s_j^2 = 1$ and the fact that $dQ_2 = dQ_1 \cdot \max_{j' \notin I_k} s_{j'}^2 \lesssim Q_1$ due to the assumption $d \max_{j' \notin I_k} s_{j'}^2 = O(1)$. Then, we can see that

$$\|A^{-1}\tilde{\phi}\|^2 = c_A^{-2}\tilde{\phi}^\top B^{-1}(G + \hat{\lambda}_0 I)^{-1}B^{-1}\tilde{\phi} = c_A^{-2}\zeta_s^\top (G + \hat{\lambda}_0 I)^{-1}\zeta_s,$$

for $\zeta_s = (s_1^{-1}, \ldots, s_k^{-1}, 0, \ldots, 0)^\top$. Now, let

$$G + \hat{\lambda}_0 = \begin{pmatrix} G_{[1,1]} & G_{[1,2]} \\ G_{[2,1]} & G_{[2,2]} \end{pmatrix}.$$

Then, we can see that

$$(G + \hat{\lambda}_0 I)^{-1} = \begin{pmatrix} (G_{[1,1]} - G_{[1,2]}G_{[2,2]}^{-1}G_{[2,1]})^{-1} & * \\ * & * \end{pmatrix}.$$

We know that $\|G_{[2,2]}^{-1}\|_{\mathrm{op}} \leq \|(\hat{\lambda}_0 I)^{-1}\|_{\mathrm{op}} = Q_2^{-1}$ and $\|G_{[1,2]}\|_{\mathrm{op}} \lesssim C_0\sqrt{k(d-k)Q_1 Q_2/(kd)}$ by the same argument as Eq. (16) and the assumption $d \max_{j' \notin I_k} s_{j'}^2 = O(1)$. Hence, we can see that

$$G_{[1,1]} - G_{[1,2]}G_{[2,2]}^{-1}G_{[2,1]} \gtrsim Q_1 I - \mathcal{O}(C_0^2(k(d-k)Q_1 Q_2/(kd))/Q_2)I \gtrsim [Q_1 - \mathcal{O}(C_0^2 Q_1)]I,$$

by Eq. (17). Therefore, by taking $C_0$ sufficiently small, we have that

$$(G_{[1,1]} - G_{[1,2]}G_{[2,2]}^{-1}G_{[2,1]})^{-1} \lesssim Q_1^{-1}I.$$

Therefore, we finally arrive at

$$\|A^{-1}\tilde{\phi}\|^2 \leq c_A^{-2}\|\zeta_s\|^2\|(G_{[1,1]} - G_{[1,2]}G_{[2,2]}^{-1}G_{[2,1]})^{-1}\|_{\mathrm{op}}$$

$$\lesssim kQ_1\left(\max_{j'} s_{j'}^2\right) \cdot k\left(\min_{j' \in I_k} s_{j'}^2\right)^{-1} \cdot Q_1^{-1} = k^2 \frac{\max_{j' \in [d]} s_{j'}^2}{\min_{j' \in I_k} s_{j'}^2}.$$

## C  KERNEL LOWER BOUND

In this section, we derive the kernel lower bound for the $k$-parity classification problem (Example 2) in the spiked covariance setting. Our proof is divided into two steps. First, we translate the event when prediction fails into when the value of $|f_\beta(z)|$ is away from zero. We combine the proof for 2-parity (Wei et al., 2019) and an additional observation that $K(z, z^i)$ have $d^{-(1-\alpha)k^* - (k-k^*)}$ correlation to $y$, to get the tighter bound for general higher order parities than that in Wei et al. (2019). Then, we show that the probability of that event is evaluated by the the smallest eigenvalue of some kernel matrix defined in Lemma 3. Finally, we apply the lower bound of the smallest eigenvalue using the technique of Misiakiewicz (2022).

Note that, we do not need to prove Theorem 3 for $1 - \frac{1}{k^*} < \alpha \leq 1$, because $\lfloor (1-\alpha)k^* \rfloor = 0$ holds for such $\alpha$. Thus in the following we assume $1 - \frac{1}{k^*} < \alpha \leq 1$, hence $1 - \alpha > 0$.

**Lemma 1.** *For $n \leq d^{(1-\alpha)k^* + (k-k^*)}$, with probability $1 - \exp(-\Omega(d))$ over the random draws of the training sample, we have*

$$\mathbb{P}_{z \sim P_Z}[f_\beta(z)y < 0] \gtrsim \mathbb{P}_{z \sim P_Z}\left[|f_\beta(z)| \geq \frac{c}{d^{(1-\alpha)k^* + (k-k^*)}}\sum_{i=1}^n |\beta_i|\right] - 1/d,$$

*where $c$ is a constant depending on $k$ and $\{\gamma_l\}_l$.*

*Proof.* Consider randomly drawn $z_{k+1:d}$, which we fix in the following. Suppose $f_\beta(z)y(z) \geq 0$ for all choices of $z_{1:k}$ and $|f_\beta(z)| \gtrsim \frac{c}{d^{(1-\alpha)k}}\sum_{i=1}^n |\beta_i|$ for some $z_{1:k}$ for the sake of contradiction (with high probability). Then, consider the average of $K(z, z^i)y$ over the choices of $z_{1:k}$ as follows:

$$\mathbb{E}_{z_{1:k}}\left[K(z, z^i)y(z)\big| z_{k+1:d}\right] = \mathbb{E}_{z_{1:k}}\left[\sum_{l=0}^\infty \alpha_l\left(\frac{z^\top z^i}{d}\right)^l y(z)\bigg| z_{k+1:d}\right]$$

$$= \sum_{l=k}^{\infty} \gamma_l \mathbb{E}_{z_{1:k}} \left[ \left( \frac{z^\top z^i}{d} \right)^l \prod_{j'=1}^{k} z_{j'} \,\middle|\, z_{k+1:d} \right] \tag{18}$$

Let us evaluate $\mathbb{E}_{z_{1:k}}[(\frac{z^\top z^i}{d})^l \prod_{j'=1}^{k} z_{j'} | z_{k+1:d}]$. For $k \leq l \leq \lceil \frac{2((1-\alpha)k^* + (k-k^*))}{1-\alpha} \rceil$, we expand $(\frac{z^\top z^i}{d})^l = (\sum_{i=j}^{d} \frac{z_j z_j^i}{d})^l$ to obtain

$$\mathbb{E}_{z_{1:k}} \left[ \left( \frac{z^\top z^i}{d} \right)^l \prod_{j'=1}^{k} z_{j'} \,\middle|\, z_{k+1:d} \right] \leq \underbrace{\sum_{l'=k}^{l} {}_l C_{l'} k^{l'} (d-k)^{l-l'} \left( \frac{d^\alpha}{d} \right)^{k^*} \left( \frac{1}{d} \right)^{k-k^*} \left( \frac{1}{d} \right)^{l-l'}}_{\text{terms containing each } z_1, \cdots, z_k \text{ at least once}}$$

$$\lesssim d^{-(1-\alpha)k^* - (k-k^*)}.$$

For $l \geq \lceil \frac{2((1-\alpha)k^* + (k-k^*))}{1-\alpha} \rceil + 1$, we have $|\frac{z^\top z^i}{d}| \lesssim d^{-(1-\alpha)/2} \sqrt{\log d}$ with probability $1 - 1/d^{(1-\alpha)k + (k-k^*)+1}$ over the choice of $z_{k+1:d}$, and therefore $\sum_{l=2k+1}^{\infty} \mathbb{E}_{z_{1:k}}[|\frac{z^\top z^i}{d}|^l | z_{k+1:d}] \lesssim d^{-(1-\alpha)k^* - (k-k^*)}$. By using them for (18), we have

$$\mathbb{E}_{z_{1:k}} \left[ K(z, z^i) y(z) \,\middle|\, z_{k+1:d} \right] = \mathbb{E}_{z_{1:k}} \left[ \sum_{l=0}^{\infty} \alpha_l \left( \frac{z^\top z^i}{d} \right)^l y(z) \,\middle|\, z_{k+1:d} \right] \lesssim d^{-(1-\alpha)k^* - (k-k^*)}$$

for randomly drawn $z_{k+1:d}$, with probability more than $1 - 1/d^{(1-\alpha)k^* + (k-k^*)+1}$. Therefore,

$$\mathbb{E}_{z_{1:k}} \left[ f_\beta(z) y(z) \,\middle|\, z_{k+1:d} \right] = \mathbb{E}_{z_{1:k}} \left[ \sum_i \beta_i K(z, z^i) y(z) \,\middle|\, z_{k+1:d} \right] \tag{19}$$

$$\lesssim \frac{1}{d^{(1-\alpha)k^* + (k-k^*)}} \sum_{i=1}^{n} |\beta_i| \tag{20}$$

with probability more than $1 - 1/d$.

On the other hand, if $f_\beta(z) y(z) \geq 0$ for all $z_{1:k}$ and $|f_\beta(z)| \gtrsim \frac{c}{d^{(1-\alpha)k^* + (k-k^*)}} \sum_{i=1}^{n} |\beta_i|$ for some $z_{1:k}$, we have

$$\mathbb{E}_{z_{1:k}} \left[ f_\beta(z) y(z) \,\middle|\, z_{k+1:d} \right] = \frac{1}{2^k} \sum_{z_{1:k}} f_\beta(z) y(z) \geq \frac{1}{2^k} \cdot \frac{c}{d^{(1-\alpha)k^* + (k-k^*)}} \sum_{i=1}^{n} |\beta_i|. \tag{21}$$

By comparing (20) and (21), we have the contradiction for more than $1 - 1/d$ probability of the choice of $z_{k+1:d}$ by taking $c$ sufficiently large. Therefore, if $|f_\beta(z)| \gtrsim \frac{c}{d^{(1-\alpha)k^* + (k-k^*)}} \sum_{i=1}^{n} |\beta_i|$ for some $z_{1:k}$, there exists some $z_{1:k}$ that yields $f_\beta(z) y < 0$, for $z_{k+1:d}$ that is drawn with probability more than $1 - 1/d$, which yields the conclusion. $\square$

Now we evaluate the probability $\mathbb{P}_{z \sim P_Z}[|f_\beta(z)| \geq \frac{c}{d^{(1-\alpha)k^* + (k-k^*)}} \sum_{i=1}^{n} |\beta_i|]$. Since $f_\beta(z)$ can have very high order terms, make the following approximation.

**Lemma 2.** *Let us define $g_1 \colon [-1, 1] \to \mathbb{R}$ as*

$$g_1(t) = \sum_{l=0}^{\lceil \frac{2((1-\alpha)k^* + (k-k^*))}{1-\alpha} \rceil} \gamma_l t^l.$$

*Suppose $n \leq d^{(1-\alpha)k^* + (k-k^*)}$. Then,*

$$\mathbb{P}_{z \sim P_Z} \left[ \exists i \in [n], \left| K(z, z^i) - g_1 \left( \frac{z^\top z^i}{d} \right) \right| \leq d^{-(1-\alpha)k^* - (k-k^*)} \right] \geq 1 - 1/d.$$

*Proof.* First, we note

$$\left| K(z, z^i) - g_1 \left( \frac{z^\top z^i}{d} \right) \right| = \left| \sum_{l=0}^{\infty} \alpha_l \left( \frac{z^\top z^i}{d} \right) - \sum_{l=0}^{\lceil \frac{2((1-\alpha)k^* + (k-k^*))}{1-\alpha} \rceil} \gamma_l \left( \frac{z^\top z^i}{d} \right) \right|$$

$$= \sum_{\lceil \frac{2((1-\alpha)k^*+(k-k^*))}{1-\alpha} \rceil + 1}^{\infty} \gamma_l \left| \frac{z^\top z^i}{d} \right|. \tag{22}$$

With probability $1 - 1/d^{(1-\alpha)k^*+(k-k^*)+1}$, $\left| \frac{z^\top z^i}{d} \right| \lesssim d^{-(1-\alpha)/2}\sqrt{\log d}$. This means that (22) is

bounded by $\lesssim \left( \frac{\log d}{d^{1-\alpha}} \right)^{(\lceil \frac{2((1-\alpha)k^*+(k-k^*))}{1-\alpha} \rceil + 1)/2} \leq d^{-(1-\alpha)k^*+(k-k^*)}$ for a sufficiently large $d$. By

taking the uniform bound over all $i$, we get the assertion. $\qquad\square$

Owing to this lemma, it suffices to bound $\mathbb{P}_{z \sim P_Z}[|\sum_{i=1}^n \beta_i g_1(\frac{z^\top z^i}{d})| \geq \frac{c}{d^{(1-\alpha)k^*+(k-k^*)}} \sum_{i=1}^n |\beta_i|]$ by $\Omega(1)$, because

$$\mathbb{P}_{z \sim P_Z} \left[ |f_\beta(z)| \geq \frac{c}{d^{(1-\alpha)k^*+(k-k^*)}} \sum_{i=1}^n |\beta_i| \right]$$

$$\geq \mathbb{P}_{z \sim P_Z} \left[ \left| \sum_{i=1}^n \beta_i g_1 \left( \frac{z^\top z^i}{d} \right) \right| \geq \frac{c+1}{d^{(1-\alpha)k^*+(k-k^*)}} \sum_{i=1}^n |\beta_i| \right] - 1/d.$$

For this, we lower bound the second moment, which captures the variation of $f_\beta$.

**Lemma 3.** *Suppose $a_l$ are all positive and define $K_2 \in \mathbb{R}^{n \times n}$ as*

$$(K_2)_{i,j} = \sum_{l=0}^k \left( \frac{z^i_{k+1:d}{}^\top z^j_{k+1:d}}{d-k} \right)^l.$$

*Then, for sufficiently large $d$, we have*

$$\mathbb{E}_z \left[ \left( \sum_{i=1}^n \beta_i g_1 \left( \frac{z^\top z^i}{d} \right) \right)^2 \right] \gtrsim d^{-\lfloor (1-\alpha)k^* \rfloor + (k-k^*)} \beta^\top K_2 \beta.$$

The proof requires several auxiliary lemmas as follows. We defer the proofs of them after the proof of Lemma 3.

**Lemma 4.** *For any integers $p, g \geq 0$,*

$$\mathbb{E}_z \left[ \left( \sum_{i=1}^n \beta_i (z^\top z^i)^p \right) \left( \sum_{i=1}^n \beta_i (z^\top z^i)^q \right) \right]$$

$$\geq \mathbb{E}_{z_{k+1:d}} \left[ \left( \sum_{i=1}^n \beta_i (z^\top_{k+1:d} z^i_{k+1:d})^p \right) \left( \sum_{i=1}^n \beta_i (z^\top_{k+1:d} z^i_{k+1:d})^q \right) \right] \geq 0$$

**Lemma 5.** *Let $z^i, z^j \in \{-1, 1\}^d$, $z \in \{-1, 1\}^d$ be a vector sampled uniformly from the hypercube, and let $l$ be any integer. Then, we can expand the expectation as*

$$\mathbb{E}_z \left[ \left( \frac{z^\top z^i}{d} \right)^l \left( \frac{z^\top z^j}{d} \right)^l \right] = \sum_{l'=0}^l d^{-l} c_{d,l,l'} \left( \frac{z^i{}^\top z^j}{d} \right)^{l'}.$$

*Furthermore, for sufficiently large $d$, $c_{d,l,l'} \geq 0$ and especially $c_{d,l,l} = (l!)^2$.*

*Proof of Lemma 3.* Let us first expand the target:

$$\mathbb{E}_z \left[ \left( \sum_{i=1}^n \beta_i g_1 \left( \frac{z^\top z^i}{d} \right) \right)^2 \right]$$

$$= \mathbb{E}_z \left[ \left( \sum_{i=1}^n \beta_i \sum_{l=0}^{\lceil \frac{2((1-\alpha)k^*+(k-k^*))}{1-\alpha} \rceil} \gamma_l \left( \frac{z^\top z^i}{d} \right)^l \right)^2 \right]$$

$$= \mathbb{E}_z \left[ \left( \sum_{l=0}^{\lceil \frac{2((1-\alpha)k^* + (k-k^*))}{1-\alpha} \rceil} \gamma_l \sum_{i=1}^{n} \beta_i \left( \frac{z^\top z^i}{d} \right)^l \right)^2 \right]$$

$$= \sum_{0 \le l_1, l_2 \le \lceil \frac{2((1-\alpha)k^* + (k-k^*))}{1-\alpha} \rceil} \gamma_{l_1} \gamma_{l_2} \mathbb{E}_z \left[ \left( \sum_{i=1}^{n} \beta_i \left( \frac{z^\top z^i}{d} \right)^{l_1} \right) \left( \sum_{i=1}^{n} \beta_i \left( \frac{z^\top z^i}{d} \right)^{l_2} \right) \right] \quad (23)$$

From Lemma 4 and $\gamma_{l_1}, \gamma_{l_2} > 0$, each term is non-negative and (23) is lower bounded by

$$\sum_{l=0}^{\lceil \frac{2((1-\alpha)k^* + (k-k^*))}{1-\alpha} \rceil} \gamma_l^2 \mathbb{E}_{z_{k+1:d}} \left[ \left( \sum_{i=1}^{n} \beta_i \left( \frac{z_{k+1:d}^\top z_{k+1:d}^i}{d} \right)^l \right)^2 \right]$$

$$\gtrsim \sum_{l=0}^{\lceil \frac{2((1-\alpha)k^* + (k-k^*))}{1-\alpha} \rceil} \gamma_l^2 \mathbb{E}_{z_{k+1:d}} \left[ \left( \sum_{i=1}^{n} \beta_i \left( \frac{z_{k+1:d}^\top z_{k+1:d}^i}{d-k} \right)^l \right)^2 \right]. \quad (24)$$

Let us define a matrix $K_1 \in \mathbb{R}^{n \times n}$ so that (24) is equal to $\beta^\top K_1 \beta$. For that, we define

$$(K_1)_{i,j} = \sum_{l=0}^{\lceil \frac{2((1-\alpha)k^* + (k-k^*))}{1-\alpha} \rceil} \gamma_l^2 \mathbb{E}_{z_{k+1:d}} \left[ \left( \frac{z_{k+1:d}^\top z_{k+1:d}^i}{d-k} \right)^l \left( \frac{z_{k+1:d}^\top z_{k+1:d}^j}{d-k} \right)^l \right].$$

According to Lemma 5,

$$(K_1)_{i,j} = \sum_{l=0}^{\lceil \frac{2((1-\alpha)k^* + (k-k^*))}{1-\alpha} \rceil} \gamma_l^2 \sum_{l'=0}^{l} (d-k)^{-l} c_{d-k,l,l'} \left( \frac{z_{k+1:d}^i \,^\top z_{k+1:d}^j}{d} \right)^{l'}$$

$$= \sum_{l=0}^{\lceil \frac{2((1-\alpha)k^* + (k-k^*))}{1-\alpha} \rceil} \left( \sum_{l''=l}^{\lceil \frac{2((1-\alpha)k^* + (k-k^*))}{1-\alpha} \rceil} \gamma_{l''}^2 (d-k)^{-l''} c_{d-k,l'',l} \right) \left( \frac{z_{k+1:d}^i \,^\top z_{k+1:d}^j}{d-k} \right)^l.$$

Because $c_{d-k,l'',l} \ge 0$ and $c_{d-k,l,l} = (l!)^2$, $(d-k)^{-l} c_l := \left( \sum_{l''=l}^{2k} \gamma_{l''}^2 (d-k)^{-l''} c_{d-k,l'',l} \right) \gtrsim d^{-l}$ holds. Thus, we have $(d-k)^{-l} c_l \ge d^{-\lfloor (1-\alpha)k^* \rfloor + (k-k^*)} c$ for all $l \le \lfloor (1-\alpha)k^* \rfloor + (k-k^*)$ for sufficiently small $c$, and by defining $K_2, K_3 \in \mathbb{R}^{n \times n}$ as

$$(K_2)_{i,j} = \sum_{l=0}^{\lfloor (1-\alpha)k^* \rfloor + (k-k^*)} \left( \frac{z_{k+1:d}^i \,^\top z^j}{d-k} \right)^l$$

$$(K_3)_{i,j} = \sum_{l=0}^{\lfloor (1-\alpha)k^* \rfloor + k - k^*} \left( (d-k)^{-l} c_l - d^{-(\lfloor (1-\alpha)k^* \rfloor + k - k^*)} c \right) \left( \frac{z_{k+1:d}^i \,^\top z_{k+1:d}^j}{d-k} \right)^l$$

$$+ \sum_{l=\lfloor (1-\alpha)k^* \rfloor + k - k^* + 1}^{\lceil \frac{2((1-\alpha)k^* + (k-k^*))}{1-\alpha} \rceil} (d-k)^{-l} c_l \left( \frac{z_{k+1:d}^i \,^\top z_{k+1:d}^j}{d-k} \right)^l,$$

we have $K_1 = c d^{-\lfloor (1-\alpha)k^* \rfloor + k - k^*} K_2 + K_3$. Moreover, $K_3$ is positive semi-definite because $K_3$ is written as a sum of polynomial kernels with positive coefficients. Thus, we can lower bound $\beta^\top K_1 \beta$ by $d^{-\lfloor (1-\alpha)k^* \rfloor - (k-k^*)} \beta^\top K_2 \beta$ (up to a constant factor). $\qquad \square$

*Proof of Lemma 4.* The proof idea comes from Lemma B.9 of Wei et al. (2019). For a set $S \subseteq [k]$, we let $z^S = \prod_{i=1}^{k} z_i$, and for a set $T \subseteq [d] \setminus [k]$, we let $z^T = \prod_{i=1}^{k} z_i$. Expand $(z^\top z^i)^p$ as

$$(z^\top z^i)^p = \left( \sum_{j=1}^{d} z_j z_j^i \right)^p = \sum_{S,T} C_{|S|,|T|,p} z^S z^T (z^i)^S (z^i)^T,$$

where $c_{|S|,|T|,p} \geq 0$ depends only on $|S|$, $T$, and $p$ due to symmetry. Also, we let

$$(z_{k+1:d}^\top z_{k+1:d}^i)^p = \sum_T \bar{C}_{|T|,p} z_{k+1:d}^S z_{k+1:d}^T (z_{k+1:d}^i)^S (z_{k+1:d}^i)^T.$$

Note that $C_{0,|T|,p} \geq \bar{C}_{|T|,p} \geq 0$, because $C_{0,|T|,p}$ considers the case where $z_i (i \in [k])$ is multiplied for even number of times.

As a basic fact in Boolean analysis, we have $\mathbb{E}_z[z^S z^T z^{S'} z^{T'}] = 0$ unless $S = S'$ and $T = T'$. Therefore,

$$\mathbb{E}_z\left[\left(\sum_{i=1}^n \beta_i (z^\top z^i)^p\right)\left(\sum_{i=1}^n \beta_i (z^\top z^i)^q\right)\right]$$

$$= \mathbb{E}_z\left[\left(\sum_{i=1}^n \beta_i \sum_{S,T} C_{|S|,|T|,p} z^S z^T (z^i)^S (z^i)^T\right)\left(\sum_{i=1}^n \beta_i \sum_{S,T} C_{|S|,|T|,q} z^S z^T (z^i)^S (z^i)^T\right)\right]$$

$$= \sum_{S,T} \mathbb{E}_z\left[\left(\sum_{i=1}^n \beta_i C_{|S|,|T|,p} (z^i)^S (z^i)^T\right)\left(\sum_{i=1}^n \beta_i C_{|S|,|T|,q} z^S z^T (z^i)^S (z^i)^T\right)\right]$$

$$= \sum_{S,T} d^{2|S|\alpha} C_{|S|,|T|,p} C_{|S|,|T|,q} \left(\sum_{i=1}^n \beta_i\right)^2$$

$$\geq \sum_T C_{0,|T|,p} C_{0,|T|,q} \left(\sum_{i=1}^n \beta_i\right)^2 \tag{25}$$

Where we used $C_{|S|,|T|,p}, C_{|S|,|T|,q} \geq 0$. On the other hand,

$$\mathbb{E}_{z_{k+1:d}}\left[\left(\sum_{i=1}^n \beta_i (z_{k+1:d}^\top z_{k+1:d}^i)^p\right)\left(\sum_{i=1}^n \beta_i (z_{k+1:d}^\top z_{k+1:d}^i)^q\right)\right] = \sum_T \bar{C}_{|T|,p} \bar{C}_{|T|,q} \left(\sum_{i=1}^n \beta_i\right)^2 \geq 0. \tag{26}$$

Because $c_{|S|,|T|,p} \geq \bar{C}_{T,p}$ and $c_{|S|,|T|,q} \geq \bar{C}_{T,q}$, comparing (25) and (26) yields

$$\mathbb{E}_z\left[\left(\sum_{i=1}^n \beta_i (z^\top z^i)^p\right)\left(\sum_{i=1}^n \beta_i (z^\top z^i)^q\right)\right]$$

$$\geq \mathbb{E}_{z_{k+1:d}}\left[\left(\sum_{i=1}^n \beta_i (z_{k+1:d}^\top z_{k+1:d}^i)^p\right)\left(\sum_{i=1}^n \beta_i (z_{k+1:d}^\top z_{k+1:d}^i)^q\right)\right] \geq 0,$$

which concludes the proof. □

*Proof of Lemma 5.* LHS is determined by how many coordinates are different between $z^i$ and $z^j$, which is captured by $z^{i\top} z^j$. Thus, LHS is the polynomial of $z^{i\top} z^j$. Moreover, its degree is at most $l$ because the degrees of $z^\top z^i$ and $z^\top z^j$ are at most $l$ in LHS. Thus the LHS can be written as $\sum_{l'=0}^l c_{d,l,l'} \left(\frac{z^{i\top} z^j}{d^2}\right)^{l'}$. Note that, when $l$ is even, LHS is invariant to the replacement $z^j \mapsto -z^j$, and therefore $c_{d,l,l'} = 0$ for odd $l'$. On the other hand, when $l$ is odd, $c_{d,l,l'} = 0$ for even $l'$.

Let us evaluate $c_{d,l,l'}$. By multiplying $d^l$ on both sides, we have

$$\mathbb{E}_z\left[\left(\frac{z^\top z^i}{\sqrt{d}}\right)^l \left(\frac{z^\top z^j}{\sqrt{d}}\right)^l\right] = \sum_{l'=0}^l c_{d,l,l'} \left(\frac{z^{i\top} z^j}{d}\right)^{l'}.$$

By taking $d \to \infty$ (while fixing the angle $\frac{z^{i\top} z^j}{d}$), LHS will converge into

$$\mathbb{E}_g\left[\left(\frac{g^\top z^i}{\sqrt{d}}\right)^l \left(\frac{g^\top z^j}{\sqrt{d}}\right)^l\right], \tag{27}$$

here $g$ follows $\mathbb{S}^{d-1}(\sqrt{d})$. Now consider the Hermite expansion of $t^l = \sum_{l'=0}^{l} c_{l,l'} He_{l'}(t)$. If $l$ is even, $c_{l,l'} = \frac{1}{2^{\frac{l-l'}{2}}(\frac{l-l'}{2})! l'!} > 0$ for even $l'$ and $c_{l,l'} = 0$ for odd $l'$. If $l$ is odd, $c_{l,l'} = \frac{1}{2^{\frac{l-l'}{2}}(\frac{l-l'}{2})! l'!} > 0$ for odd $l'$ and $c_{l,l'} = 0$ for even $l'$. By using these Hermite coefficients, (27) is equal to

$$\sum_{l'=0}^{l'} c_{l,l'}^2 \left( \frac{z^{i\top} z^j}{d} \right)^{l'}.$$

Note that, as a function of the angle $\frac{z^{i\top} z^j}{d} \in [-1, 1]$, the convergence is uniform. Therefore, we get

$$d^{-l} c_{d,l,l'} \to c_{l,l'}^2 \quad (d \to \infty)$$

for all $l$ and $l'$. When $c_{l,l'}^2 = 0$, $c_{d,l,l'} = 0$ for all $d$ as we saw above. When $c_{l,l'}^2 > 0$, there exists $d$ such that $c_{d',l,l'} > 0$ for all $d' \geq d$. Therefore, for sufficiently large $d$, we have $c_{d,l,l'} \geq 0$. Moreover, by direct calculation, $c_{d,l,l} = (l!)^2$. $\qquad\square$

After obtained Lemma 3 we would like to bound $d^{-\lfloor (1-\alpha)k \rfloor} \beta^\top K_2 \beta$. For this, we use the lower bound the smallest eigenvalue of $K_2$.

Let $K_{(d)}$ $(d = 1, 2, \cdots)$ be a sequence of inner-product kernels with $K_{(d)}(z, z') = h_{(d)}(\frac{z^\top z'}{d})$. Consider the case when each $K_{(d)}$ is associated with the same kernel function $h \colon [-1, 1] \to \mathbb{R}$, so that $h_{(d)} = h$ holds for all $z, z' \in \{-1, 1\}^d$. The following Lemma requires $h$ is a degree-$k$ polynomial and its coefficients are positive for all degrees. Note that $K_2$ satisfies these conditions.

**Lemma 6** (Misiakiewicz (2022)). *Assume the following conditions hold:*

(a) $h^{(k')}(0) > 0$ *for* $k' = 0, \cdots, k - 1$.

(b) $h^{(k)}(0) > 0$.

(c) $h(\cdot)$ *is $k$-times differentiable.*

*Now fix $\delta > 0$ arbitrarily, and assume that $d \gg 1$ and $n \lesssim d^k e^{-a_d \sqrt{\log d}}$ for some $\{a_d\}$ with $a_d \to \infty (d \to \infty)$. Given $n$ i.i.d. sample $\{z^i\}_{i=1}^n$ from $P_Z$, we construct a kernel matrix $K \in \mathbb{R}^{n \times n}$ as $(K_{(d)})_{i,j} = h(\frac{z^{i\top} z^j}{d})$. Then, the kernel matrix $K_{(d)}$ can be decomposed into two positive semi-definite kernel $K_{>k-1}$ and $K_{\leq k-1}$, and the spectrum of $K_{>k-1}$ is bounded by*

$$\mathbb{E}_{\{z^i\}_{i=1}^n} \left[ \| K_{>k-1} - h^{(k)}(0) I \|_{\text{op}}^2 \right] \to 0 \quad (d \to \infty).$$

*Proof.* See Section 3.2 of Misiakiewicz (2022), where we take $\kappa = k - \delta$. $\qquad\square$

Therefore, for any fixed $\delta > 0$ and $d \gg 1$ and $n \lesssim d^{\lfloor (1-\alpha)k^* \rfloor + (k-k^*) - \delta}$, all the assumptions are satisfied for $K_2$ with $k = \lfloor (1-\alpha)k^* \rfloor + (k - k^*)$ (if we regard $K_2$ as a kernel in $\mathbb{R}^{d-k} \times \mathbb{R}^{d-k}$). Note that we can take $a_d = (\log d)^{\frac{1}{4}}$ so that and $d^k e^{-a_d \sqrt{\log d}} \gtrsim d^{k-\delta}$. Then, the smallest eigenvalue of $K_{>k-1}$ is lower bounded by $\Omega(1)$ with probability at least 0.99 over the randomly drawn sample, for sufficiently large $d$. This immediately implies that the smallest eigenvalue of $K_2$ is bounded by $\Omega(1)$ with probability at least 0.99.

Now we finalize the proof of Theorem 3.

*Proof of Theorem 3.* According to Lemmas 3 and 6, for all choices of $\beta$, with probability at least 0.99 over the randomly drawn sample, we have

$$\mathbb{E}_z \left[ \left( \sum_{i=1}^n \beta_i g_1 \left( \frac{z^\top z^i}{d} \right) \right)^2 \right] \gtrsim d^{-\lfloor (1-\alpha)k^* \rfloor - (k-k^*)} \sum_{i=1}^n \beta_i^2 \tag{28}$$

$$\geq \frac{1}{d^{\lfloor (1-\alpha)k^* \rfloor + (k-k^*)} n} \left( \sum_{i=1}^n |\beta_i| \right)^2 \tag{29}$$

$$\gtrsim \frac{1}{d^{2\lfloor(1-\alpha)k^*\rfloor+2(k-k^*)-\delta}}\left(\sum_{i=1}^{n}|\beta_i|\right)^2. \tag{30}$$

Because $g_1$ is a degree-$2k$ polynomial, Bonami's Lemma (e.g., Theorem 9.21 of (O'Donnell, 2014)) yields

$$\mathbb{E}_z\left[\left(\sum_{i=1}^{n}\beta_i g_1\left(\frac{z^\top z^i}{d}\right)\right)^4\right] \geq \frac{1}{(2k-1)^{4k}}\mathbb{E}_z\left[\left(\sum_{i=1}^{n}\beta_i g_1\left(\frac{z^\top z^i}{d}\right)\right)^2\right]^2$$

As a result, the Paley–Zygmund inequality (see Theorem 9.4 of (O'Donnell, 2014)) yields

$$\mathbb{P}_z\left[\left|\sum_{i=1}^{n}\beta_i g_1\left(\frac{z^\top z^i}{d}\right)\right| \geq t\mathbb{E}_z\left[\left(\sum_{i=1}^{n}\beta_i g_1\left(\frac{z^\top z^i}{d}\right)\right)^2\right]^{\frac{1}{2}}\right] \geq \frac{(1-t^2)^2}{(2k-1)^{4k}} \tag{31}$$

for all $0 \leq t \leq 1$.

Combining (28) and (31), with probability 0.99 over the sample, we have

$$\left|\sum_{i=1}^{n}\beta_i g_1\left(\frac{z^\top z^i}{d}\right)\right| \gtrsim \frac{1}{d^{\lfloor(1-\alpha)k^*\rfloor+(k-k^*)-\delta/2}}\sum_{i=1}^{n}|\beta_i|.$$

with probability $\Omega(1)$ over the choice of $z$. By taking sufficiently large $d$, $\frac{1}{d^{\lfloor(1-\alpha)k^*\rfloor+(k-k^*)-\delta/2}}$ is larger than $\frac{1+c}{d^{\lfloor(1-\alpha)k^*\rfloor+(k-k^*)}}$ ($c$ is a constant from Lemma 1). Thus, using Lemma 2, we get

$$\mathbb{P}_{z\sim P_Z}\left[|f_\beta(z)| \geq \frac{c}{d^{\lfloor(1-\alpha)k^*\rfloor+(k-k^*)}}\sum_{i=1}^{n}|\beta_i|\right] \gtrsim 1-1/d.$$

Now we apply Lemma 1 and finally obtain

$$\mathbb{P}_{z\sim P_Z}\left[f_\beta(z)y < 0\right] \gtrsim 1-2/d,$$

which concludes the proof. □

## D  DETAILS OF THE EXPERIMENT

We describe the experiment settings for Figure 1. We considered an anisotropic $d$-dimensional 3-sparse parity problem (Example 2): $y = z_1 z_2 z_3$, $s_1 = s_2 = s_3 = \alpha/\sqrt{d}$, and $s_4 = \cdots = 1/\sqrt{d}$. Here $\alpha$ controls the alignment of the distribution to the feature, or the signal-to-noise ratio. We fixed the dimension $d$ to 300, and varied $n$ and $\alpha$. We trained the neural network (2) with $\bar{R} = 15$. Specifically, we employed the width $N = 2000$ as a finite neuron approximation, and initialized neurons so that each of them followed the standard normal distribution (and thus the network was rotation invariant at the initialization). By using the logistic loss, we updated the network by the discretized MLFD (6) by setting $\eta = 0.25$, $\lambda_1 = 0.1$, and $\lambda = 0.1\alpha^2/d$ (fixed during the training) by following Corollary 1, until $T = 10000$. We ran the experiment 5 times with different seeds and plotted the mean for each $n$ and $\alpha$.

