# OpenReview forum: "Improved statistical and computational complexity of the mean-field Langevin dynamics under structured data"
_ICLR.cc/2024/Conference — ICLR 2024 poster_

### Official Review · Reviewer_Gjnf · 2023-10-31

**Soundness:** 4 excellent
**Presentation:** 4 excellent
**Contribution:** 3 good
**Rating:** 6
**Confidence:** 3

**Summary:**

The authors address how gradient-based feature learning interacts with anistropic input data. They do this by studying the sparse parity problem where feature coordinates have varying magnitudes. To this end, they analyze the learning complexity of the mean-field Langevin dynamics, which describes the noisy gradient descent update on two-layer neural networks. They show one can use the anisotropy of the data to improve the statistical complexity (i.e. sample size) and computational complexity (i.e. network width) of the mean-field Langevin dynamics. This improvement is found when the main directions of the anisotropic input data aligns with the support of the target function. They also provide a method using coordinate transformations determined by the gradient covariance to show that the computational complexity can be improved by exploiting the anisotropy.

**Strengths:**

- The results seem novel and interesting. Namely they show how one can utilize the anisotropy of the input data to improve learning for a specific setting.
- The result of coordinate transformations which leverages anisotropy could be of interest for practical neural network training.
- The implication of a tradeoff between statistical and computational complexity also seems interesting.
- The paper is clearly written and the results clearly stated.

**Weaknesses:**

- In my opinion, the claim that "anisotropy helps" seems too strong, a bit specific to the problem in the paper, and not necessarily a general statement that can be made for neural networks based on the assumptions and results of the paper. It would be great if the authors could better motivate a connection between this problem and more general neural networks with real-world datasets.

- The authors state on page 2, under "Feature learning under structured data", that

  > in certain regression settings with low-dimensional target, structured data with a spiked covariance structure can improve the performance of both kernel methods and optimized NNs (Ghorbani et al., 2020; Ba et al., 2023; Mousavi-Hosseini et al., 2023; Suzuki et al., 2023b). However, these regression analyses do not directly translate to the binary classification setting which the k-parity problem belongs to.

  This implies that the intuition "anisotropy helps" is perhaps already observed in the literature, which may weaken the impact of the paper. If the authors could expand on why the $k$-parity problem is worth studying separately, that would really help motivate things.

**Questions:**

- One page 7, under Corollary 1, the authors state:
  > On the other hand, if the input covariance is anisotropic so that ... then the value of $R$ becomes dimension-free: $R = O(k^2 log(k)^2 )$."

  Can similar results apply to real-world datasets?

- How applicable is the coordinate transformation used in the paper, to conventional neural network training for real-world datasets?

- For equation (2), we're using a smooth activation function (tanh). Can the results apply for nonsmooth activation functions, namely ReLU networks? i.e. can the intuition "anisotropy helps" also apply for ReLU networks?

---

> ### Author Response · Authors · 2023-11-18
>
> We thank the reviewer for the positive evaluation and helpful feedback. We address the technical comments below.
>
> **Connection to more general neural networks with real-world datasets**
>
> First, we note that our primary goal is to develop theory for representation learning – to understand when it is beneficial in terms of statistical and computational complexity, and how it interacts with different aspects of the learning problem / algorithm, such as structure of the data and target function, model architecture, and optimization algorithm. More specifically, the focus of this work is to rigorously examine the interplay between feature learning and the anisotropic structure of input data, which, as remarked in the introduction, is relatively underexplored especially in the context of classification problems. While our theoretical setting is idealized, it is motivated from practical observations that real-world data exhibits low-dimensional structure (see below), and we believe that our analysis serves as an important step towards a concrete understanding of feature learning in more complicated and realistic settings.
>
> On the other hand, our problem setting is motivated by empirical observations on realistic data. Real-world data, including image and text, can be high-dimensional, yet neural networks often efficiently learn from data and avoid the “curse of dimensionality”, This success can be attributed to the fact that real-world data exhibits some underlying structure, and intrinsic low dimensionality is considered as one of the prominent factors. Specifically, low intrinsic dimensionality has the following two aspects.
> (1) *Low-dimensionality of ground truth (target function)*. This means that not all directions of the input features are important for predicting $y$.
> (2) *Anisotropy of input data*. This means that the features already contain low-dimensional structures despite the large ambient dimension.
> The $k$-sparse parity classification problem is a classical example of low-dimensional function acting on high-dimensional data, and most prior theoretical studies for this problem setting considered isotropic input data, which accounts for (1) but not (2). By introducing this “generalized” version of the sparse parity problem on anisotropic input, we aim to theoretically study the interplay between structured (anisotropic) data and the efficiency of feature learning via gradient descent. We show that neural networks trained by gradient-based algorithms indeed exploit such low-dimensional structure, as evident in the improved statistical and computational complexity.
> As for application to real-world data, our Theorem 1 suggests that neural networks can leverage low-dimensional structure to improve generalization performance, whereas Theorem 2 suggests that the structure of target function can be revealed by the gradient covariance matrix. We believe that these insights can be transferred to practical settings at a high level.
>
> **Difference from prior results on anisotropic data**
>
> We have revised the Introduction to highlight the difference from prior works and new technical challenges. We make the following remarks.
> 1. Classification and regression problems have fundamentally different structures: it is possible that a predictor achieves vanishing classification error but has large regression loss. Concretely, the following aspects of our analysis do not translate from the regression setting:
> * For the neural network results, we exploited properties of the logistic loss, and more importantly, margin conditions on the optimal classifier. Such analysis does not follow from reducing the classification problem to regression (e.g., with the squared loss).
> * For the kernel lower bound, establishing a lower bound on the classification error is much more challenging than on the regression loss, because it is clear that a kernel model can perfectly predict the sign of the labels $y$ but do not achieve small squared error.
> 2. Regarding the optimization dynamics, our analysis differs significantly from prior results (Ghorbani et al., 2020; Ba et al., 2023; Mousavi-Hosseini et al., 2023). (Ghorbani et al., 2020) did not provide any optimization guarantees for neural networks. (Ba et al., 2023; Mousavi-Hosseini et al., 2023) considered a narrow neural network in the so-called “rotation” regime, where the statistical complexity is governed by the *information exponent* of the ground truth. Roughly speaking, to learn a degree-$k$ parity function in this regime, the sample complexity may scale with $d^{\Theta(k)}$. In contrast, we focused on neural networks in the mean-field regime, which allows us to “decouple” the degree from the exponent of the dimension dependence, albeit at a cost of higher computational complexity (Theorem 1). Finally, we realize that (Suzuki et al., 2023b) does not handle the anisotropic setting – we have removed this citation in the paragraph on structured data.

---

> > ### Author Response · Authors · 2023-11-18
> >
> > **"Why the $k$-parity problem is worth studying separately"**
> >
> > The $k$-sparse parity classification problem often serves as a testbed for theory of representation learning, and has been extensively studied in the isotropic setting  – see references in the Introduction section. The reason is twofold: (i) The target function only depends on a few coordinates of the input features, and hence we expect that the trained neural network representation can “zoom-in” to the relevant k-dimensional subspace. (ii) For methods that do not exploit such low-dimensional structure, such as neural networks in the lazy (NTK) regime, we expect the sample complexity to be inferior to the feature learning alternative – this provides a simple setting to illustrate the benefit of representation learning.
> >
> > **"How applicable is the coordinate transformation used in the paper, to conventional neural network training for real-world datasets?"**
> >
> > The coordinate transform can be implemented in practical neural networks, since the gradient covariance estimation can be done by standard auto-differentiation analogous to the computation of second-order statistics for preconditioned gradient update. We intuitively expect such transformation to be beneficial when the batch size is sufficiently large. It is also possible that the covariance of SGD noise is also implicitly amplifying certain informative directions, thus achieving a similar effect as explicit coordinate transformation; we leave such investigation to future work.
> >
> > **"Can the results apply for nonsmooth activation functions, namely ReLU networks?"**
> >
> > Smooth activation is a technical condition required in the current convergence analysis of the mean-field Langevin dynamics, and is also present in many existing convergence guarantees for mean-field neural networks. But since we can approximate a ReLU arbitrarily well with smooth activation, we expect that similar intuition is still valid for practical networks with ReLU activation.
> >
> > We would be happy to clarify any further concerns/questions in the discussion period.

---

> > > ### Comment · Reviewer_Gjnf · 2023-11-20
> > >
> > > I thank the authors for their nicely written reply. I also see they have updated the paper, and I find these updates to be good revisions. I have also read the other reviews.
> > >
> > > The authors have mostly addressed my concerns. In general, I remain reserved about whether findings from studying just the $k$-sparse parity problem can generalize to more settings, and specifically to real-world settings. As mentioned in my review, this can be alleviated if the authors had more experiments closer to real-world situations.
> > >
> > > For now, I am inclined to keep my score, but recognize that the authors have made good clarifications in the discussion, and positive updates to their paper.

---

### Official Review · Reviewer_e822 · 2023-11-07

**Soundness:** 3 good
**Presentation:** 3 good
**Contribution:** 2 fair
**Rating:** 6
**Confidence:** 2

**Summary:**

This work investigates mean-field Langevin dynamics on structured anisotropic input data with a k-sparse parity target function. It provides results for specific 2-layer neural networks around both statistical and computational complexity. In particular, it proves discrete-time and finite-width learning guarantees thus extending on the results of previous work. These results are verified empirically on synthetic data.

**Strengths:**

Disclaimer: I have not checked the validity of the proofs in the appendix and, therefore, cannot comment on the correctness of the results.

* This work is generally well written.
* The related work seems to be well addressed (although I am not familiar with the literature).
* The contributions are quite clearly laid out in the introduction.
* Section 2 (and 3) does a reasonable job of laying out the problem setting.

**Weaknesses:**

* I am not well-positioned to comment on the significance of the results within this research area. However, it is not clear to me why we should be interested in this formal setting beyond the fact that some previous works have considered it. The assumptions seem highly contrived and no attempt is made to link them to any practical task in a meaningful way. Could the authors explain why progress in this research direction is worth pursuing? I would argue that if this work is being submitted to a broad conference such as ICLR, more effort should be made to broaden its appeal by explaining why its setting is relevant.

**Questions:**

* Given the very niche nature of this paper, I'm not sure that ICLR is the best venue for this work. Given that (I would imagine) this work would be of interest to quite a small subcommunity, would it not be more suitable to submit to a more specific venue rather than a generalist conference like ICLR? This is not to speak negatively about this work, but it seems that this topic is less generally relevant in its nature and may not be best suited to this large-scale style of conference.

* Could the authors provide a definition of anisotropic input data? I don't think this was clearly defined. Does it exactly refer to the setting defined mathematically at the beginning of Sec 1.1?

* Why can the existing results for the regression setting not be directly applied in the binary classification setting by converting it to a regression problem?

---

> ### Author Response · Authors · 2023-11-18
>
> We thank the reviewer for the helpful feedback.
>
> The reviewer’s main concern is regarding the relevance of our theoretical findings to the ICLR community. We motivate our problem setting from the following three perspectives.
>
> **1. Why do we care about the theory of representation (feature) learning?**
>
> It is clear that representation learning is a central topic of ICLR, as indicated by the title of the conference. We argue that it is imperative to develop theory for representation learning – to understand when it is beneficial in terms of statistical and computational complexity, and how it interacts with different aspects of the learning problem / algorithm, such as structure of the data and target function, model architecture, and optimization algorithm. The focus of this work is to rigorously examine the interplay between feature learning and the anisotropic structure of input data, which, as remarked in the introduction, is relatively underexplored especially in the context of classification problems. While our theoretical setting is idealized, it is motivated from practical observations that real-world data exhibits low-dimensional structure (see point below), and we believe that our analysis serves as an important step towards a concrete understanding of feature learning in more complicated and realistic settings, and is certainly of interest to the ICLR community. In fact, there are at least 5 papers accepted to ICLR 2023 that theoretically analyzed the feature learning dynamics under similar idealized settings: [Mousavi et al.](https://openreview.net/pdf?id=6taykzqcPD), [Bordelon and Pehlevan](https://openreview.net/pdf?id=nZ2NtpolC5-), [Akiyama and Suzuki](https://openreview.net/pdf?id=6doXHqwMayf), [Telgarsky](https://openreview.net/pdf?id=swEskiem99), [Tian](https://openreview.net/pdf?id=s130rTE3U_X).
>
> **2. Why is the sparse parity problem worth studying?**
>
> The $k$-sparse parity classification problem is a classical example of low-dimensional function acting on high-dimensional data. This function often serves as a testbed for theory of representation learning, and has been extensively studied in the isotropic setting  – see references in the Introduction section. The reason is twofold: (i) The target function only depends on a few coordinates of the input features, and hence we expect that the trained neural network representation can “zoom-in” to the relevant k-dimensional subspace. (ii) For methods that do not exploit such low-dimensional structure, such as neural networks in the lazy (NTK) regime, we expect the sample complexity to be inferior to the feature learning alternative – this provides a simple setting to illustrate the benefit of representation learning.
>
> **3. What new insights do we gain from introducing anisotropy?**
>
> Real-world data, including image and text, can be high-dimensional, yet neural networks often efficiently learn from data and avoid the “curse of dimensionality”, This success can be attributed to the fact that real-world data exhibits some underlying structure, and intrinsic low dimensionality is considered as one of the prominent factors. Specifically, low intrinsic dimensionality has the following two aspects (see [Pope et al.](https://arxiv.org/pdf/2104.08894.pdf) for example).
> (1) *Low-dimensionality of ground truth (target function)*. This means that not all directions of the input features are important for predicting $y$.
> (2) *Anisotropy of input data*. This means that the features already contain low-dimensional structures despite the large ambient dimension.
>
> As noted in the previous point, prior theoretical studies on the isotropic parity problem accounted for (1) but not (2). By introducing this “generalized” version of the sparse parity problem on anisotropic input, we aim to theoretically study the interplay between structured (anisotropic) data and the efficiency of feature learning via gradient descent. We show that neural networks trained by gradient-based algorithms indeed exploit such low-dimensional structure, as evident in the improved statistical and computational complexity. Similar problem setups and motivations have appeared in [Ghorbani et al. (2020)](https://proceedings.neurips.cc/paper/2020/hash/a9df2255ad642b923d95503b9a7958d8-Abstract.html) and [Ba et al. (2023)](https://openreview.net/forum?id=HlIAoCHDWW) but for the regression setting.
>
> We have revised the Introduction to highlight the motivation of our problem setting and the relevance of our theoretical results.

---

> > ### Author Response · Authors · 2023-11-18
> >
> > Now we address the additional questions in the review.
> >
> > **"Why can the existing results for the regression setting not be directly applied in the binary classification setting by converting it to a regression problem?"**
> >
> > Note that classification and regression problems have fundamentally different structures: it is possible that a predictor achieves vanishing classification error but has large regression loss. Concretely, the following aspects of our analysis cannot be obtained by converting the classification problem into regression:
> > * For the neural network results, we exploited properties of the logistic loss, and more importantly, margin conditions on the optimal classifier. Such analysis does not follow from reducing the classification problem to regression (e.g., with the squared loss).
> > * For the kernel lower bound, establishing a lower bound on the classification error is much more challenging than on the regression loss, because it is clear that a kernel model can perfectly predict the sign of the labels $y$ but do not achieve small squared error.
> >
> >
> > **"Could the authors provide a definition of anisotropic input data?"**
> >
> > We follow the standard definition of isotropy/anisotropy, that is, a random vector is anisotropic if its covariance is not identity.
> >
> > We would be happy to clarify any further concerns/questions in the discussion period.

---

> > ### Comment · Reviewer_e822 · 2023-11-20
> >
> > I am grateful to the authors for their rebuttal. My primary issue with this work (and related works) is with respect to the claim that "our analysis serves as an important step towards a concrete understanding of feature learning in more complicated and realistic settings". It is not clear to me that this is true -- the path of understanding settings such as the one in this work and translating these findings into practical scenarios is something that appears less and less plausible to me over time. I hope future work will make a greater attempt to demonstrate this claimed value in some form.
> >
> > However, I appreciate that there may be some interest in the research community in this work which was certainly well presented. I do not have any strong reason to reject this paper based on technical complaints and therefore I raise my score to 6.

---

### Official Review · Reviewer_obgQ · 2023-11-08

**Soundness:** 3 good
**Presentation:** 3 good
**Contribution:** 3 good
**Rating:** 8
**Confidence:** 3

**Summary:**

In this work, the authors study the statistical and computational complexity of mean-field Langevin dynamics (MFLD) with anisotropic input data. In particular, they show that both complexities can be improved when prominent directions of the anisotropic input data align with the support of the target function.

**Strengths:**

This paper is technically solid and study an important problem. MFLD is a recent framework to help us understand the behavior of two-layer nonlinear NN, particularly relevant to the isotropic $k$-parity problem. Extending the isotropic setting to the anisotropic setting is definitely interesting to study, and the authors have established learning guarantees for two-layer nonlinear NN for this anisotropic setting.

**Weaknesses:**

It appears that the whole work assumes that the matrix $A$ is known a priori or prespecified. To me, this assumption might be too strong and make the problem of more theoretical interest than very practically relevant, although it seems to me that the motivation of studying anisotropic data arises from the case of realistic datasets as mentioned in the abstract.

**Questions:**

If we have real data while $A$ is generally unavailable or can only be estimated from data, what can we say about this case with the results of this paper?

I also wonder if the problem can be tackled with a preconditioned version of MFLD, instead of performing coordinate transforms on the input.

Typo:
- page 6, after Proposition 2: otherwise, we “still”
- page 6, after Proposition 3: Under this “condition”

---

> ### Author Response · Authors · 2023-11-18
>
> We thank the reviewer for the positive evaluation and insightful comments. We have revised the manuscript and corrected the typos you pointed out. We address the technical points below.
>
> **"The whole work assumes that the matrix $A$ is known a priori or prespecified"**
>
> We make the following remarks.
> * In the data generating process, we assume a fixed transformation $A$ (which is not known to the learning algorithm). This problem setting is motivated by the empirical observation that real-world features exhibit low-dimensional structure, which is not captured by many prior works on feature learning assuming isotropic input. By introducing this “generalized” version of the sparse parity problem on anisotropic input, we aim to theoretically study the interplay between structured data and the efficiency of feature learning via gradient descent. To further illustrate the benefit of feature learning in the anisotropic setting, in the revised manuscript, we introduced a stronger kernel lower bound that applies to a wide range of (spiked) anisotropic data.
> * We emphasize that the transformation $A$ is not given to the learning algorithm; instead, the algorithm only has access to i.i.d. observations from the data generating process, and our goal is to characterize how anisotropy (controlled by the matrix $A$) enhances the performance of MFLD. Similar problem setups and motivations have appeared in [Ghorbani et al. (2020)](https://proceedings.neurips.cc/paper/2020/hash/a9df2255ad642b923d95503b9a7958d8-Abstract.html) and [Ba et al. (2023)](https://openreview.net/forum?id=HlIAoCHDWW) for the regression setting.
> * As for implications for real-world data, since the learning algorithm does not know $A$ a priori, intuitively speaking, what Theorem 1 suggests is that neural networks can leverage low-dimensional structure to improve generalization performance, whereas Theorem 2 suggests that the structure of target function can be revealed by the gradient covariance matrix. We believe that these insights can be transferred to real-world settings.
>
> **"I also wonder if the problem can be tackled with a preconditioned version of MFLD"**
>
> Thank you for the interesting suggestion. At a high level, there is indeed a parallel between the coordinate transformation and preconditioning. However, we note that naively preconditioning the entire update yields the stationary distribution unchanged (though the optimization speed might improve), whereas our coordinate transform on the input feature yields a different stationary distribution on the parameter space, which enables us to establish a better LSI constant and generalization. It is an interesting question of whether we should apply the preconditioner to the gradient of the loss, the regularization, or the Gaussian noise separately, as here different design choices would result in different stationary distributions of MFLD. We intend to investigate this perspective in future work.
>
> We would be happy to clarify any further concerns/questions in the discussion period.

---

> > ### Comment · Reviewer_obgQ · 2023-11-23
> >
> > Thanks the authors for the response and some comments on my suggestion. This seems to be an interesting direction to pursue and I hope to see the future work which definitely make the problem concerned more practically relevant.

---

### Author Response · Authors · 2023-11-18
**Summary of revision**

Dear Reviewers and Area Chair,

We appreciate your continuing time and effort to provide detailed comments on our paper. To best respond to the reviews, we revised the manuscript with additional clarifying content as suggested by the reviewers. Below is a short summary of the updates.
* Following the reviewers’ suggestions, in the Introduction section we further motivated our problem setting, including $(i)$ how the anisotropic data model relates to real-world data, and $(ii)$ how our results differ from prior analyses of feature learning under structured data.
* We fixed the typos pointed out by the reviewers, and improved the presentation of our main results. For instance, in Table 1 we adopted the more intuitive notion of *effective dimension* $d_{\text{eff}}$ from [Ghorbani et al. (2020)](https://proceedings.neurips.cc/paper/2020/hash/a9df2255ad642b923d95503b9a7958d8-Abstract.html).
* We reorganized Section 4 and included additional discussion/interpretation of our theoretical findings. We also strengthened the kernel lower bound, which is now presented separately in Section 5 (see point below).
* We extended the kernel classification lower bound to the spiked covariance setting (which includes the previous isotropic lower bound as a special case). This provides a more comprehensive comparison between kernel methods (non-adaptive) and mean-field neural networks (adaptive).

Please refer to our detailed response to each reviewer where we address all individual comments. We note that since the main text has been updated to improve clarity and readability, some numbering of theorems has changed in the revised version; we refer to the updated numbering in the rebuttal.

---

### Meta-Review · Area_Chair_i5Q8 · 2023-12-11

**Metareview:**

The authors present a theoretical analysis of learning in two layer neural networks that generalizes previous work which only considered isotropic input data distributions, and show that an anisotropic input distribution can lead to faster learning. The technical quality of work is high, but there were concerns that the topic might be somewhat niche, and that the work was only an incremental advance over the previous work on the topic. Nevertheless, none of the reviewers found these objections serious enough to reject the paper. I concur and recommend the paper be accepted.

**Justification For Why Not Higher Score:**

This is addressed in the metareview.

**Justification For Why Not Lower Score:**

This is addressed in the metareview.

---

### Decision · Program_Chairs · 2024-01-16

Accept (poster)